# DDGA: Dirichlet Distributional Gradient Aggregation for Transferable Vision-Language Adversarial Attacks

**Yiwei You** [1]   **Jiaan Wei** [1]   **Zan Chen** [1]   **Bo Wang** [1]

## Abstract

Vision-Language Models (VLMs) achieve remarkable performance on multimodal tasks but remain highly vulnerable to adversarial examples, making transferable attacks essential for realistic robustness evaluation. Recent Adversarial Evolution Triangle (AET) methods improve transferability by interpolating over a simplex formed by clean and historical adversarial samples, yet rely on finite random sampling to approximate effective perturbation distributions, which is unstable under limited budgets. In this paper, we propose Dirichlet Distributional Gradient Aggregation (DDGA), a distribution-aware adversarial attack framework that explicitly models and optimizes perturbations over the AET simplex. DDGA parameterizes simplex mixing weights with a learnable Dirichlet policy and optimizes the expected adversarial objective via policy gradient, replacing heuristic sampling with principled distributional optimization. Moreover, we exploit the closed-form covariance of the learned distribution to construct orthogonal perturbations that enhance gradient diversity. Extensive experiments on image-text retrieval and image captioning demonstrate that DDGA consistently outperforms state-of-the-art transfer-based attacks across multiple VLM architectures.

## 1. Introduction

Vision–Language Models (VLMs) have emerged as a fundamental paradigm for multimodal intelligence, enabling joint reasoning over visual and textual information (Zhang et al., 2024). Large-scale pre-trained models, such as CLIP (Radford et al., 2021), demonstrate remarkable zero-shot gener-

alization and achieve strong performance on various downstream tasks, including image–text retrieval (ITR) (Khan & Fu, 2021), image caption (IC) (Hu et al., 2022), and visual question answering (VQA) (Hartsock & Rasool, 2024). These advances have positioned VLMs as core components in real-world AI systems (Ghosh et al., 2024).

Despite their impressive capabilities, recent studies show that VLMs are highly vulnerable to adversarial samples crafted in either single or multiple modalities (Gu et al., 2023). Carefully designed adversarial examples can significantly disrupt cross-modal alignment and mislead model predictions, raising serious concerns about the reliability and safety of VLMs in practical deployment (Díaz-Rodríguez et al., 2023). Consequently, developing stronger adversarial attack methods is essential, not only for stress-testing model robustness, but also for guiding the design of more effective defenses (Ying et al., 2025; Wang et al., 2025).

A central challenge in adversarial evaluation of VLMs is **attack transferability** (Hu et al., 2025). Modern VLMs are typically large-scale and deployed as black-box services, making white-box attacks costly or impractical (Chen et al., 2025). Therefore, generating adversarial examples from source model that reliably transfer across unseen target models is essential for realistic threat assessment. To enhance transferability, Adversarial Evolution Triangle (AET) were recently proposed (Gao et al., 2024; Jia et al., 2025). AET constructs a simplex spanned by the clean input and historical adversarial samples, and generates new perturbations through stochastic interpolation within this simplex. By reusing past adversarial information, AET aims to explore a broader and more diverse perturbation space than single-step gradient updates, thereby identifying directions that generalize better across models (Jia et al., 2025).

Specifically, as a representative AET-based baseline, DRA uniformly samples multiple adversarial candidates from the AET simplex, selects the most adversarial sample, and further applies data augmentation to improve diversity along the optimization trajectory (Gao et al., 2024). While such sampling-based strategies are empirically effective and conceptually simple, they rely on finite stochastic sampling to approximate the unknown distribution of effective perturbations within the simplex. Under practical query or iteration

---

[1]School of Artificial Intelligence and Data Science, University of International Business and Economics, Beijing, China. Correspondence to: Bo Wang <wangbo@uibe.edu.cn>.

*Proceedings of the 43rd International Conference on Machine Learning*, Seoul, South Korea. PMLR 306, 2026. Copyright 2026 by the author(s).

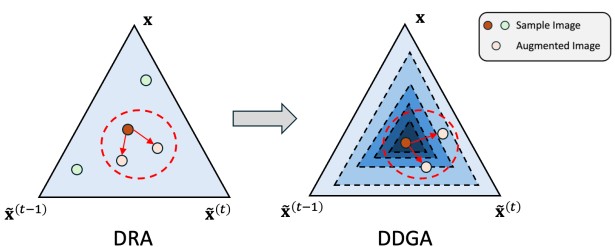

*Figure 1.* Illustration of DRA versus DDGA. **Left**: DRA relies on finite random interpolation within the adversarial simplex, yielding unstable and incomplete exploration of perturbations. **Right**: DDGA learns a continuous perturbation distribution over the simplex, illustrated by probability density contours, and optimizes the expected perturbation via policy gradient, resulting in more stable and transferable adversarial directions.

budgets, this approximation can be inefficient and exhibit high variance, potentially missing informative perturbation directions in high-dimensional spaces (Dong et al., 2025), which highlights the absence of an explicit mechanism for modeling and optimizing the perturbation distribution itself. These limitations naturally motivate a key question:

*Can we directly learn the perturbation distribution over the AET simplex, rather than relying on the demanding heuristic sampling, to generate more effective and transferable adversarial examples?*

In this paper, we propose *Dirichlet Distributional Gradient Aggregation* (DDGA), a principled framework that explicitly models and optimizes the perturbation distribution to enhance adversarial transferability. As illustrated in Figure 1, DRA generates new adversarial samples via heuristic sampling over a simplex of historical examples. In contrast, our DDGA formulates this process as a learnable distributional optimization problem: We parameterize the simplex mixing weights using a *Dirichlet distribution* and update its parameters via policy gradient based on adversarial feedback. This design enables adaptive identification of informative gradient directions rather than relying on unstable random sampling. Furthermore, we exploit the second-order structure of the learned distribution by incorporating covariance-guided orthogonal perturbations, which encourage the diversity among aggregated gradients.

By jointly optimizing visual perturbations and adversarial text generation within this distribution-aware framework, DDGA produces more consistent and transferable adversarial samples. Extensive experiments on ITR and IC demonstrate that DDGA consistently outperforms existing transfer-based attack methods across multiple VLMs. Comprehensive ablations further validate the effectiveness of each component and the practical robustness of our approach.

In summary, our main contributions are three-fold:

- We provide a theoretical analysis of AET-based attacks,

revealing that gradient aggregation is governed by the underlying perturbation distribution and highlighting an inefficiency of finite stochastic sampling.

- We propose DDGA, which explicitly learns a Dirichlet mixing distribution over the AET simplex via policy gradient and exploits its second-order structure to generate diverse and transferable adversarial perturbations.

- Extensive experiments on ITR and IC demonstrate that DDGA consistently improves cross-model and cross-task transferability over state-of-the-art baselines.

## 2. Background

### 2.1. Preliminary

Let $(\mathbf{x}, \mathbf{t}) \in \mathcal{D}$ be an image-text pair sampled from a multimodal dataset, where $\mathbf{x}$ is an image, and $\mathbf{t}$ is its textual description. A pretrained VLM is denoted as $\mathcal{F} = \{F_I, F_T\}$, where $F_I(\cdot)$ and $F_T(\cdot)$ are image and text encoders, respectively. Given an image-text pair $(\mathbf{x}, \mathbf{t})$, VLM projects both modalities into a shared embedding space and aligns them via cosine similarity (Radford et al., 2021). Specifically, for a set of candidate texts $\mathcal{T} = \{\mathbf{t}_1, \mathbf{t}_2, \ldots, \mathbf{t}_C\}$, the probability of assigning image $\mathbf{x}$ to candidate text $\mathbf{t}_c$ is computed using a softmax over similarity scores as:

$$g_c(\mathbf{x}, \mathcal{T}) = \frac{\exp(\text{sim}(F_I(\mathbf{x}), F_T(\mathbf{t}_c)))}{\sum_{i=1}^{C} \exp(\text{sim}(F_I(\mathbf{x}), F_T(\mathbf{t}_i)))}, \quad (1)$$

where $\text{sim}(\cdot, \cdot)$ denotes cosine similarity. Consider $y \in \{1, 2, \ldots, C\}$ denotes the ground-truth index of the text corresponding to image $\mathbf{x}$. In ITR task, the objective is to correctly retrieve the related text $\mathbf{t}_y$ given an image $\mathbf{x}$.

Adversarial attacks on VLMs aim to disrupt the semantic alignment between visual and textual representations by introducing imperceptible perturbations to both modalities, while satisfying predefined perturbation constraints. Let $\mathbb{B}(\mathbf{x}, \epsilon_I)$ and $\mathbb{B}(\mathbf{t}, \epsilon_T)$ denote the $\ell_\infty$-norm balls centered at $\mathbf{x}$ and $\mathbf{t}$ with radii $\epsilon_I$ and $\epsilon_T$, respectively. Multimodal adversarial examples $(\tilde{\mathbf{x}}, \tilde{\mathbf{t}})$ are generated by maximizing the cross-entropy loss $\ell_{\text{CE}}$, yielding the following optimization problem (Jia et al., 2025):

$$\underset{\mathbf{x} \in \mathbb{B}(\mathbf{x}, \epsilon_I), \mathbf{t} \in \mathbb{B}(\mathbf{t}, \epsilon_T)}{\text{argmax}} \ell_{\text{CE}}(g(\mathbf{x}, \mathcal{T}), y), \quad \text{for } \mathbf{t} \in \mathcal{T}. \quad (2)$$

Prior work commonly adopts iterative Projected Gradient Descent (PGD) to construct adversarial images under $\ell_\infty$ constraints. Starting from an initial perturbation $\tilde{\mathbf{x}}^{(0)} = \mathbf{x} + 0.001 \cdot \mathcal{N}(\mathbf{0}, \mathbf{I})$, the update at step $t$ is given by:

$$\tilde{\mathbf{x}}^{(t+1)} = \underset{\mathbb{B}(\mathbf{x}, \epsilon_I)}{\Pi} \left[ \tilde{\mathbf{x}}^{(t)} + \eta \cdot \text{sign}\left( \nabla_{\tilde{\mathbf{x}}^{(t)}} \ell(g(\tilde{\mathbf{x}}^{(t)}, \mathcal{T}), y) \right) \right], \quad (3)$$

where $\eta$ denotes the step size and $\Pi_{\mathbb{B}(\mathbf{x}, \epsilon_I)}(\cdot)$ projects the adversarial image back onto the feasible perturbation set.

## 2.2. Transferability-Enhanced Adversarial Attacks

To improve adversarial transferability, SGA (Lu et al., 2023) incorporates image-space data augmentation into PGD process. Let $\tilde{\mathbf{x}}^{(t)}$ be adversarial image obtained at the $t$-th step. At step $t+1$, SGA applies image augmentations, e.g., resizing to multiple scales, to $\tilde{\mathbf{x}}^{(t)}$, yielding a set of $M$ augmented samples $\{\tilde{\mathbf{x}}_1^{(t)}, \tilde{\mathbf{x}}_2^{(t)}, \ldots, \tilde{\mathbf{x}}_M^{(t)}\}$. The adversarial update is then computed by aggregating the gradients over all augmented samples as:

$$\tilde{\mathbf{x}}^{(t+1)} = \prod_{\mathbb{B}(\mathbf{x}, \epsilon_I)} \left[ \tilde{\mathbf{x}}^{(t)} + \eta \cdot \text{sign}\left(\nabla_{\tilde{\mathbf{x}}^{(t)}} \sum_{m=1}^{M} \ell(g(\tilde{\mathbf{x}}_m^{(t)}, \mathcal{T}), y))\right)\right]. \quad (4)$$

To further enhance transferability, Jia et al. proposed sampling from *adversarial evolution triangles* (AET). Specifically, at step $t$, DRA samples $N$ instances from the triangle $\Delta(\mathbf{x}, \tilde{\mathbf{x}}^{(t-1)}, \tilde{\mathbf{x}}^{(t)})$, forming a candidate set $S^{(t)} = \{s_1^{(t)}, s_2^{(t)}, \ldots, s_N^{(t)}\}$. In detail, each sample is constructed as a convex combination (Gao et al., 2024; Jia et al., 2025):

$$s_n^{(t)} = \lambda \cdot \mathbf{x} + \beta \cdot \tilde{\mathbf{x}}^{(t-1)} + \gamma \cdot \tilde{\mathbf{x}}^{(t)}, \quad \text{s.t. } \lambda + \beta + \gamma = 1, \quad (5)$$

where $\lambda$, $\beta$, and $\gamma$ are weights used in the convex combination. DRA evaluates all samples in $S^{(t)}$ using the cross-entropy loss $\ell_{\text{CE}}$ and selects the most adversarial instance. The selected sample is then used as the input to the SGA procedure, after which the next-step adversarial image $\tilde{\mathbf{x}}^{(t+1)}$ is generated using the PGD update in Eq. (4).

Such AET-based methods, typically construct adversarial updates by sampling multiple candidates within the simplex and selecting the most adversarial one. Although effective in practice, this strategy relies on a finite number of samples to approximate the underlying distribution of adversarially effective perturbations within the simplex. As a result, the selected sample may fail to capture informative directions when the sampling budget is limited, and the optimal sampling pattern can vary substantially across different inputs.

## 3. Method

In this section, we provide a theoretical perspective on AET-based attacks and indicate that their effectiveness is fundamentally governed by *underlying perturbation distribution* rather than by individual sampled perturbations, which motivates us to explicitly model and optimize the perturbation distribution within the simplex.

### 3.1. Second-Order Perspective on Gradient Aggregation

We first analyze the local behavior of the adversarial loss under small input perturbations. For simplicity, we focus on the image modality and omit the text input.

Let $g(\tilde{\mathbf{x}}, \mathcal{T}) = g(\tilde{\mathbf{x}})$ denote the output probability vector of the VLM, and consider the cross-entropy loss $\ell_{\text{CE}}(g(\tilde{\mathbf{x}}), y) = -\log g_y(\mathbf{x} + \boldsymbol{\delta})$, where $\boldsymbol{\delta}$ denotes the perturbation applied to the clean image $\mathbf{x}$. Then, define the Jacobian and Hessian Matrices of $g_y$ with respect to the image input as:

$$J_y(\mathbf{x}) \triangleq \nabla_{\mathbf{x}} g_y(\mathbf{x}), \qquad H_y(\mathbf{x}) \triangleq \nabla_{\mathbf{x}}^2 g_y(\mathbf{x}). \quad (6)$$

**Lemma 3.1.** *For sufficiently small perturbations $\boldsymbol{\delta}$, the gradient of the cross-entropy loss with respect to $\boldsymbol{\delta}$ admits the following second-order approximation:*

$$\nabla_{\boldsymbol{\delta}} \ell_{\text{CE}}(g(\mathbf{x}+\boldsymbol{\delta}), y) \approx -\frac{J_y(\mathbf{x})}{g_y(\mathbf{x})} - \left[\frac{H_y(\mathbf{x})}{g_y(\mathbf{x})} - \frac{J_y(\mathbf{x}) J_y(\mathbf{x})^\top}{g_y(\mathbf{x})^2}\right]\boldsymbol{\delta}. \quad (7)$$

Lemma 3.1 reveals that, within a local neighborhood of the clean image, the loss gradient is an *affine function* of the perturbation $\boldsymbol{\delta}$. The first term provides a baseline ascent direction, while the second term characterizes how different perturbations linearly modulate this direction. Consequently, the contribution of multiple perturbed inputs to gradient aggregation can be fully described by the first-order statistics of the perturbation distribution.

**Theorem 3.2.** *Under the conditions of Lemma 3.1, let $\boldsymbol{\delta}$ be sampled from a perturbation distribution $\Delta_{\mathbf{x}}$ with mean $\bar{\boldsymbol{\delta}} = \mathbb{E}_{\boldsymbol{\delta} \sim \Delta_{\mathbf{x}}}[\boldsymbol{\delta}]$. Then, we have*

$$\mathbb{E}_{\boldsymbol{\delta} \sim \Delta_{\mathbf{x}}}\left[\nabla_{\boldsymbol{\delta}} \ell_{\text{CE}}(g(\mathbf{x} + \boldsymbol{\delta}), y)\right] \approx \nabla_{\boldsymbol{\delta}} \ell_{\text{CE}}(g(\mathbf{x} + \boldsymbol{\delta}), y)|_{\boldsymbol{\delta} = \bar{\boldsymbol{\delta}}}. \quad (8)$$

The proof of Lemma 3.1 and Theorem 3.2 can be found in Appendix A. Theorem 3.2 establishes a crucial implication: *To second order, gradient aggregation over a perturbation distribution provides no additional directional information beyond that captured by its mean.* Equivalently, averaging gradients computed at multiple perturbations is approximately equivalent to evaluating a single gradient at the expected perturbation. This result suggests that the effectiveness of AET-based sampling strategies, is fundamentally determined by the underlying distribution of adversarial perturbations rather than individual sampled points (Gao et al., 2024). Consequently, if the perturbation distribution within the AET $\Delta(\mathbf{x}, \tilde{\mathbf{x}}^{(t-1)}, \tilde{\mathbf{x}}^{(t)})$ can be explicitly modeled, the attack gradient induced by its expected perturbation can be directly optimized. This observation motivates directly optimizing the perturbation distribution itself, which forms the basis of our proposed DDGA framework.

### 3.2. Dirichlet Policy Gradient for Adversarial Mixing

Motivated by Theorem 3.2, we aim to directly optimize the *expected perturbation* used for gradient updates, rather than heuristically aggregating gradients from multiple samples. To this end, we parameterize the perturbation mixing process as a learnable probability distribution and optimize its parameters to maximize the expected adversarial effect by employing the policy gradient (Sutton et al., 1999).

**Dirichlet Mixing Policy.** At step $t$ of PGD attack, we consider three candidate inputs: the clean image $\mathbf{x}$, the previous adversarial example $\tilde{\mathbf{x}}^{(t-1)}$, and the current adversarial example $\tilde{\mathbf{x}}^{(t)}$. We first sample a mixing weight vector

$$\mathbf{w} = (w_1, w_2, w_3) \sim \mathrm{Dir}(\boldsymbol{\alpha}), \tag{9}$$

where $\boldsymbol{\alpha} \in \mathbb{R}_+^3$ denotes the learnable concentration parameters of the Dirichlet distribution. Then, the resulting mixed input is constructed as a convex combination:

$$\mathbf{x}_{\mathrm{mix}}(\mathbf{w}) = w_1 \cdot \mathbf{x} + w_2 \cdot \tilde{\mathbf{x}}^{(t-1)} + w_3 \cdot \tilde{\mathbf{x}}^{(t)}. \tag{10}$$

Dirichlet distribution naturally enforces simplex constraint $\sum_k w_k = 1, w_k \geq 0$, which is well-suited for modeling adaptive interpolation among multiple adversarial candidates.

**Reward Definition.** To assess the adversarial effectiveness of a sampled mixing strategy, we directly define the reward as the cross-entropy loss incurred by the mixed input:

$$R(\mathbf{w}) = \ell_{\mathrm{CE}}(g(\mathbf{x}_{\mathrm{mix}}(\mathbf{w}), \mathcal{T}), y). \tag{11}$$

A higher reward indicates stronger disruption of the image-text alignment, and hence a more aggressive adversarial mixture. We adopt this loss-based reward for its simplicity and stability. Our goal is to learn the Dirichlet parameters $\boldsymbol{\alpha}$ that maximize the expected adversarial loss:

$$J(\boldsymbol{\alpha}) = \mathbb{E}_{\mathbf{w} \sim \mathrm{Dir}(\boldsymbol{\alpha})}\big[R(\mathbf{w})\big]. \tag{12}$$

By Theorem 3.2, optimizing this expectation directly corresponds to optimizing the effective gradient direction induced by the mean perturbation of the mixing distribution.

**Policy Gradient Optimization.** We optimize $J(\boldsymbol{\alpha})$ using the score-function estimator:

$$\nabla_{\alpha_k} J(\boldsymbol{\alpha}) = \mathbb{E}_{\mathbf{w}} \left[ R(\mathbf{w}) \frac{\partial}{\partial \alpha_k} \log p(\mathbf{w} \mid \boldsymbol{\alpha}) \right]. \tag{13}$$

For the Dirichlet distribution, this yields the following unbiased estimator for each sampled $\mathbf{w}$:

$$\widehat{\nabla_{\alpha_k} J} = R(\mathbf{w}) \left( \log w_k + \psi(S) - \psi(\alpha_k) \right), \tag{14}$$

where $S = \sum_{k=1}^{3} \alpha_k$ denotes to the Dirichlet strength, and $\psi(\cdot)$ denotes the digamma function. The derivation can be found in Appendix A.

**Parameter Update.** To reduce the variance of the policy gradient estimator, we sample a set of $M$ mixing weights $\{\mathbf{w}^{(m)}\}_{m=1}^M$ independently from $\mathrm{Dir}(\boldsymbol{\alpha})$ at each iteration. The gradient is then estimated by averaging over the sampled rewards:

$$\widehat{\nabla_{\alpha_k} J} = \frac{1}{M} \sum_{m=1}^{M} R(\mathbf{w}^{(m)}) \left( \log w_k^{(m)} + \psi(S) - \psi(\alpha_k) \right). \tag{15}$$

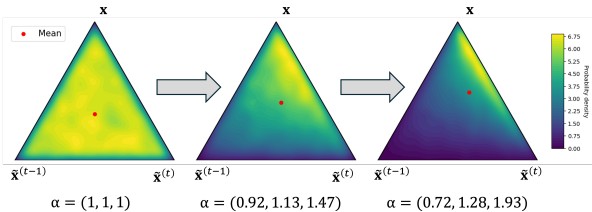

*Figure 2.* Evolution of the Dirichlet mixing policy over the simplex spanned by the clean image $\mathbf{x}$, the previous adversarial example $\tilde{\mathbf{x}}^{(t-1)}$, and the current adversarial example $\tilde{\mathbf{x}}^{(t)}$. Starting from a uniform prior, the learned policy progressively concentrates on more adversarially effective regions. The red dot denotes the mean of the Dirichlet distribution, which determines the effective perturbation used for gradient-based updates.

In the end, we update the concentration parameters using gradient ascent:

$$\alpha_k \leftarrow \alpha_k + \gamma \cdot \widehat{\nabla_{\alpha_k} J}, \tag{16}$$

where $\gamma$ denotes the learning rate of $\boldsymbol{\alpha}$.

Updating the Dirichlet parameters introduces additional computation. To keep the overall complexity comparable to prior AET-based methods, we update $\boldsymbol{\alpha}$ once per PGD iteration. In practice, this update strategy is sufficient to capture the evolving perturbation distribution while only incurring a marginal computational overhead.

**Remark 3.3.** Figure 2 provides a geometric illustration of the learned Dirichlet policy on the simplex spanned by $\Delta(\mathbf{x}, \tilde{\mathbf{x}}^{(t-1)}, \tilde{\mathbf{x}}^{(t)})$. Starting from a uniform prior, the policy progressively concentrates its mass toward adversarially effective regions, with mean shifting accordingly. Importantly, this learned distribution replaces finite-sample selection mechanism used in AET-based baselines. Instead of relying on a small number of sampled candidates to approximate the most adversarial perturbation, our method directly learns underlying perturbation distribution and uses the expected perturbation $\boldsymbol{\mu}_{\mathbf{x}} = \mathbb{E}_{\mathbf{w} \sim \mathrm{Dir}(\boldsymbol{\alpha})}[\mathbf{x}_{\mathrm{mix}}(\mathbf{w})]$, whose gradient is theoretically equivalent to aggregated gradients under second-order analysis, to guide attack update. This allows to approximate the effect of *infinitely many* perturbation samples with a single, principled gradient evaluation, leading to more stable and transferable adversarial directions.

### 3.3. Second-Order Structure-Guided Perturbation

While the expected perturbation determines the primary attack direction, i.e., Theorem 3.2, the learned Dirichlet distribution also encodes informative second-order structure. We leverage its covariance to guide structured, gradient-orthogonal exploration that improves transferability.

**Closed-Form Covariance of Dirichlet Mixing.** Given a specific learned Dirichlet distribution $\mathrm{Dir}(\boldsymbol{\alpha})$, let $\mathbf{x}_{\mathrm{mix}} =$

**Algorithm 1** Pseudo code of the proposed DDGA

---

**Input:** Image $\mathbf{x}$, texts $\mathcal{T}$, vision-language model $g = \cos(F_I(\cdot), F_T(\cdot))$, PGD steps $T$, step size $\eta$, perturbation bound $\epsilon_I, \epsilon_T$, policy learning rate $\gamma$, injection factor $\lambda$
**Initialize:** $\tilde{\mathbf{x}}^{(0)} \leftarrow \mathbf{x} + 0.001 \cdot \mathcal{N}(\mathbf{0}, \mathbf{I})$, $\boldsymbol{\alpha}^{(0)} \leftarrow (1, 1, 1)$
**for** $t = 1$ **to** $T$ **do**
    Sample $M$ weights $\{\mathbf{w}^{(m)}\}_{m=1}^{M}$ from $\mathrm{Dir}(\boldsymbol{\alpha}^{(t-1)})$
    Calculate policy gradient $\overline{\nabla}_{\alpha_k} J$ by Eq. (15)
    Update $\alpha_k^{(t)} \leftarrow \alpha_k^{(t-1)} + \gamma \cdot \widehat{\nabla_{\alpha_k} J}$
    Calculate expected perturbations $\boldsymbol{\mu}_{\mathbf{x}}$ and $\Sigma_{\mathbf{x}_{\mathrm{mix}}}$
    Employ $V$ augmentation and get $\{\boldsymbol{\mu}_{\mathbf{x}}^{(v)}\}_{v=1}^{V}$
    Calculate attack gradient $\mathbf{g}^{(t)}$ by Eq. (19)
    Calculate covariance-guided $\mathbf{v}^{\perp}$ by Eq. (21)
    $\tilde{\mathbf{x}}^{(t+1)} \leftarrow \Pi_{\mathbb{B}(\mathbf{x}, \epsilon_I)} \left( \tilde{\mathbf{x}}^{(t)} + \eta \cdot \mathrm{sign}(\mathbf{g}^{(t)} + \lambda \cdot \mathbf{v}^{\perp}) \right)$
**end for**
Generate adversarial text $\tilde{\mathbf{t}}_y$ by Eq. (23)
**Output:** Adversarial image $\tilde{\mathbf{x}}^{(T)}$, adversarial text $\tilde{\mathbf{t}}_y$

---

$\mathbf{x}_{\mathrm{mix}}(\mathbf{w}) = \mathbf{V}\mathbf{w}$ denote a Dirichlet-mixed input, where $\mathbf{V} = [\mathbf{x}, \tilde{\mathbf{x}}^{(t-1)}, \tilde{\mathbf{x}}^{(t)}] \in \mathbb{R}^{d \times 3}$ and $\mathbf{w} \sim \mathrm{Dir}(\boldsymbol{\alpha})$. The following theorem characterizes the covariance of all mixed inputs within the simplex in closed form.

**Theorem 3.4.** *Denote $S = \sum_{i=1}^{3} \alpha_i$ as the Dirichlet strength and $\boldsymbol{\mu}_{\mathbf{x}} = \mathbb{E}_{\mathbf{w} \sim \mathrm{Dir}(\boldsymbol{\alpha})}[\mathbf{x}_{\mathrm{mix}}]$ as the expected Dirichlet-mixed input. Then the covariance of $\mathbf{x}_{\mathrm{mix}}$ admits the closed-form expression*

$$\Sigma_{\mathbf{x}_{\mathrm{mix}}} = \mathbb{E}\left[(\mathbf{x}_{\mathrm{mix}} - \boldsymbol{\mu}_{\mathbf{x}})(\mathbf{x}_{\mathrm{mix}} - \boldsymbol{\mu}_{\mathbf{x}})^{\top}\right] = \mathbf{V}\mathbf{C}\mathbf{V}^{\top}, \quad (17)$$

*where $\mathbf{C} = \mathrm{Cov}(\mathbf{w}) \in \mathbb{R}^{3 \times 3}$ with entries*

$$C_{ij} = \begin{cases} \frac{\alpha_i(S - \alpha_i)}{S^2(S+1)}, & i = j, \\ -\frac{\alpha_i \alpha_j}{S^2(S+1)}, & i \neq j. \end{cases} \quad (18)$$

The proof can be found in Appendix A.3.

**Gradient-Orthogonal Covariance-Guided Exploration.**
Following Jia et al., we employ the augmentation to the expected perturbation and obtain $V$ augmented samples $\{\boldsymbol{\mu}_{\mathbf{x}}^{(1)}, \boldsymbol{\mu}_{\mathbf{x}}^{(2)}, \ldots, \boldsymbol{\mu}_{\mathbf{x}}^{(V)}\}$, and the attack gradient is computed by aggregating the gradient over all augmented samples:

$$\mathbf{g} = \frac{\sum_{v=1}^{V} \nabla_{\boldsymbol{\mu}_{\mathbf{x}}^{(v)}} \ell_{\mathrm{CE}}(g(\boldsymbol{\mu}_{\mathbf{x}}^{(v)}, \mathcal{T}), y)}{\| \sum_{v=1}^{V} \nabla_{\boldsymbol{\mu}_{\mathbf{x}}^{(v)}} \ell_{\mathrm{CE}}(g(\boldsymbol{\mu}_{\mathbf{x}}^{(v)}, \mathcal{T}), y) \|}. \quad (19)$$

Directly sampling perturbations aligned with $\mathbf{g}$ would collapse to standard PGD updates and offer limited diversity. To decouple stochastic exploration from deterministic ascent, we explicitly construct perturbations in directions orthogonal to $\mathbf{g}$. We first sample a random vector $\mathbf{r}$ from $\mathcal{N}(\mathbf{0}, \mathbf{I})$ and inject the second-order geometry of Dirichlet

mixing using the covariance matrix $\Sigma_{\mathbf{x}_{\mathrm{mix}}}$:

$$\mathbf{v} = \Sigma_{\mathbf{x}_{\mathrm{mix}}} \mathbf{r}. \quad (20)$$

To ensure that the generated perturbations do not collapse toward the deterministic PGD direction, we project $\mathbf{v}$ onto the subspace orthogonal to $\mathbf{g}$ (Jiang et al., 2026):

$$\mathbf{v}^{\perp} = \frac{\mathbf{v} - \langle \mathbf{v}, \mathbf{g} \rangle \mathbf{g}}{\| \mathbf{v} - \langle \mathbf{v}, \mathbf{g} \rangle \mathbf{g} \|}, \quad (21)$$

where orthogonal projection ensures $\langle \mathbf{v}^{\perp}, \mathbf{g} \rangle = 0$. We form the composite update direction $\mathbf{g}$ by combining the gradient direction and covariance-based orthogonal component as

$$\mathbf{g} = \mathbf{g} + \lambda \cdot \mathbf{v}^{\perp}, \quad (22)$$

where $\lambda$ controls the strength of the orthogonal injection. The resulting perturbation $\mathbf{g}$ explores high-variance directions supported by the learned Dirichlet simplex, while remaining orthogonal to the primary attack gradient. This structured stochasticity complements the PGD update and enriches the adversarial search space.

### 3.4. Adversarial Text Generation

For the text modality, we adopt BERT-attack to generate adversarial text. Given the total PGD steps $T$ and the final adversarial image $\tilde{\mathbf{x}}^{(T)}$, we generate adversarial text $\tilde{\mathbf{t}}_y$ by deviating from the features of the final augmented adversarial image (Gao et al., 2024):

$$\tilde{\mathbf{t}}_y = \underset{\tilde{\mathbf{t}}_y \in \mathbb{B}(\mathbf{t}_y, \epsilon_T)}{\mathrm{argmax}} \ell_{\mathrm{CE}}\left(g(\tilde{\mathbf{x}}^{(T)}, \mathcal{T}), y\right), \quad (23)$$

where $\mathbb{B}(\mathbf{t}_y, \epsilon_T)$ denotes the set of admissible textual perturbations under a budget $\epsilon_T$. The pseudo-code of the proposed DDGA is presented in Algorithm 1.

## 4. Related Work

**Vision-Language Models and Downstream Tasks.**
Vision-language models (VLMs) are typically pretrained on large-scale image-text pairs to learn joint visual-textual representations, and have become foundational models for various vision-language tasks (Du et al., 2022). Existing VLMs generally follow two representative architectural paradigms. The first paradigm is aligned VLMs, such as CLIP (Radford et al., 2021), which employ separate image and text encoders and align their embeddings in a shared representation space via contrastive learning. The second paradigm is fused VLMs, including ALBEF (Li et al., 2021) and TCL (Yang et al., 2022), which first extract unimodal features and then integrate them through a multimodal encoder to enable fine-grained cross-modal interactions. These pretrained representations have been widely adapted to various downstream

*Table 1.* **Cross-model comparison with state-of-the-art methods on the Flickr30K for image-text retrieval task.** We employ the $\text{CLIP}_{\text{ViT}}$ and $\text{CLIP}_{\text{CNN}}$ as the source model. We report the attack success rate (ASR) at top-1 retrieval (R@1). **Higher R@1 indicates stronger adversarial effectiveness and transferability.**

| Source | Attack | ALBEF | | TCL | | $\text{CLIP}_{\text{ViT}}$ | | $\text{CLIP}_{\text{CNN}}$ | |
|---|---|---|---|---|---|---|---|---|---|
| | | TR R@1 | IR R@1 | TR R@1 | IR R@1 | TR R@1 | IR R@1 | TR R@1 | IR R@1 |
| $\text{CLIP}_{\text{ViT}}$ | PGD | 3.13 | 6.48 | 4.43 | 8.83 | 69.33 | 84.79 | 13.03 | 17.43 |
| | BERT-Attack | 9.59 | 22.64 | 11.80 | 25.07 | 28.34 | 39.08 | 30.40 | 37.43 |
| | Sep-Attack | 7.61 | 20.58 | 10.12 | 20.74 | 76.93 | 87.44 | 29.89 | 38.32 |
| | Co-Attack | 8.55 | 20.18 | 10.01 | 21.29 | 78.53 | 87.50 | 29.50 | 38.49 |
| | SGA | 22.42 | 34.59 | 25.08 | 36.45 | 100.00 | 100.00 | 53.26 | 61.10 |
| | DRA | 27.84 | 42.84 | 27.82 | 44.60 | 100.00 | 100.00 | 64.88 | 69.50 |
| | SA-AET | 36.60 | 50.44 | 39.20 | 51.10 | 100.00 | 100.00 | **71.01** | 74.10 |
| | **DDGA** | **41.19** | **53.48** | **41.83** | **54.02** | **100.00** | **100.00** | 70.24 | **75.13** |
| $\text{CLIP}_{\text{CNN}}$ | PGD | 2.29 | 6.15 | 4.53 | 8.88 | 5.40 | 12.08 | 89.78 | 93.04 |
| | BERT-Attack | 8.86 | 23.27 | 12.33 | 25.48 | 27.12 | 37.44 | 30.40 | 40.10 |
| | Sep-Attack | 9.38 | 22.99 | 11.28 | 25.45 | 26.13 | 39.24 | 93.61 | 95.30 |
| | Co-Attack | 10.53 | 23.62 | 12.54 | 26.05 | 27.24 | 40.62 | 95.91 | 96.50 |
| | SGA | 15.64 | 28.60 | 18.02 | 33.07 | 39.02 | 51.45 | 99.87 | 99.90 |
| | DRA | 19.50 | 34.59 | 21.60 | 37.88 | 48.47 | 59.12 | 99.87 | 99.90 |
| | SA-AET | 23.98 | 38.28 | 27.29 | 41.81 | 54.11 | 64.21 | 100.00 | 99.97 |
| | **DDGA** | **27.22** | **42.42** | **31.19** | **45.17** | **60.98** | **67.59** | **100.00** | **99.97** |

tasks. Image-text retrieval evaluates cross-modal matching ability by ranking candidate images or texts (Cao et al., 2022). Image captioning aims to generate natural language descriptions conditioned on visual inputs, and is typically evaluated with captioning metrics including BLEU (Papineni et al., 2002), METEOR (Banerjee & Lavie, 2005), ROUGE (Lin, 2004), CIDEr (Vedantam et al., 2015), and SPICE (Anderson et al., 2016).

**Adversarial Attacks and Transferability in VLMs.** With the rapid adoption of VLMs in multimodal applications, their adversarial robustness has attracted increasing attention, motivating the development of both defense (Li et al., 2024; Sheng et al., 2025; Dong et al., 2025) and attack methods (Xie et al., 2025; Zhang et al., 2025). Early studies extend unimodal adversarial attacks to the multimodal setting. Sep-Attack (Lu et al., 2023) separately combines image and text perturbations, while Co-Attack (Zhang et al., 2022) exploits cross-modal interactions under white-box assumptions. To improve black-box transferability, recent methods focused on diversity-driven attack strategies. SGA (Lu et al., 2023) enhances transferability by optimizing over augmented sets of image-text pairs. Subsequent AET-based methods, such as DRA (Gao et al., 2024) and SA-AET (Jia et al., 2025), sample adversarial candidates within adversarial evolution simplex to increase perturbation diversity and improve transferable attack performance.

## 5. Experiment

**Datasets and Models.** We evaluate on two benchmarks for vision-language learning: Flickr30K (Plummer et al., 2015) and MSCOCO (Lin et al., 2014). Flickr30K contains 31,783 images with five captions per image, and MSCOCO contains 123,287 images with five captions each. Both datasets are used for image-text retrieval evaluation and cross-task adversarial transferability assessment. Adversarial examples are generated on CLIP (Radford et al., 2021) with two backbones: $\text{CLIP}_{\text{ViT}}$ (ViT-B/16 (Dosovitskiy et al., 2020)) and $\text{CLIP}_{\text{CNN}}$ (ResNet-101 (He et al., 2016)). Transferability is evaluated on multiple target models, including ALBEF (Li et al., 2021), TCL (Yang et al., 2022), $\text{CLIP}_{\text{ViT}}$, and $\text{CLIP}_{\text{CNN}}$. To assess cross-task generalization, adversarial images crafted for retrieval are further evaluated on image captioning using BLIP (Li et al., 2022). We also consider a typical multimodal large language models (MLLMs) *Qwen3-VL-4B-Instruct* to further evaluate the adversarial transferability to MLLM (Bai et al., 2025).

**Implementation details.** We adopt an $\ell_\infty$ constraint of $\epsilon_I = 8/255$, with 10 optimization iterations and step size $\alpha = 2/255$ for adversarial image. For adversarial text, we follow a single-word substitution budget $\epsilon_T = 1$ with a candidate set of 10 words (Gao et al., 2024). We employ the image augmentation strategy from SGA, resizing inputs to five scales $\{0.50, 0.75, 1.00, 1.25, 1.50\}$ via bicubic interpolation (Lu et al., 2023). For image-text retrieval, we report attack success rate (ASR) at top-1 retrieval (R@1). For image captioning, we evaluate transferability using BLEU-4, METEOR, ROUGE_L, CIDEr, and SPICE (Jia et al., 2025). See implementation details in Appendix B.

### 5.1. Main Results

**Cross-Model Adversarial Transferability.** We evaluate transferability by generating adversarial examples on the

*Table 2.* **Cross-model comparison with state-of-the-art methods on the MSCOCO image-text retrieval task.** We employ CLIP$_{\text{ViT}}$ and CLIP$_{\text{CNN}}$ as the source model. We report the attack success rate (ASR) at top-1 retrieval (R@1). **Higher R@1 indicates stronger adversarial effectiveness and transferability.**

| Source | Attack | ALBEF | | TCL | | CLIP$_{\text{ViT}}$ | | CLIP$_{\text{CNN}}$ | |
|---|---|---|---|---|---|---|---|---|---|
| | | TR R@1 | IR R@1 | TR R@1 | IR R@1 | TR R@1 | IR R@1 | TR R@1 | IR R@1 |
| CLIP$_{\text{ViT}}$ | PGD | 10.26 | 13.69 | 12.72 | 15.81 | 82.91 | 90.51 | 21.62 | 28.78 |
| | BERT-Attack | 20.34 | 29.74 | 21.08 | 29.61 | 45.06 | 51.68 | 44.54 | 53.72 |
| | Sep-Attack | 25.91 | 36.84 | 28.20 | 38.47 | 88.36 | 97.09 | 47.57 | 57.79 |
| | Co-Attack | 26.35 | 36.69 | 88.78 | 96.72 | 28.23 | 38.42 | 47.36 | 58.45 |
| | SGA | 43.75 | 51.08 | 44.05 | 51.02 | 100.00 | 100.00 | 70.66 | 75.58 |
| | DRA | 52.69 | 61.50 | 51.88 | 61.06 | 100.00 | 100.00 | 80.18 | 84.11 |
| | SA-AET | 57.64 | 66.88 | 57.30 | 65.16 | **100.00** | **100.00** | 83.98 | 86.72 |
| | **DDGA** | **66.46** | **72.53** | **64.34** | **70.67** | 99.96 | 99.96 | **87.00** | **89.76** |
| CLIP$_{\text{CNN}}$ | PGD | 8.38 | 12.73 | 11.90 | 15.68 | 13.66 | 20.62 | 92.68 | 94.71 |
| | BERT-Attack | 23.38 | 34.64 | 24.58 | 29.61 | 51.28 | 57.49 | 54.43 | 62.17 |
| | Sep-Attack | 29.13 | 40.64 | 31.40 | 42.99 | 52.23 | 59.73 | 96.16 | 97.54 |
| | Co-Attack | 29.49 | 41.50 | 31.83 | 43.44 | 53.15 | 60.15 | 97.79 | 98.54 |
| | SGA | 36.94 | 46.79 | 38.81 | 48.90 | 62.19 | 67.73 | 99.92 | **99.97** |
| | DRA | 41.40 | 52.25 | 43.62 | 54.15 | 70.43 | 74.14 | 99.80 | 99.92 |
| | SA-AET | 43.62 | 55.19 | 47.01 | 57.39 | 73.67 | 76.90 | **100.00** | 99.92 |
| | **DDGA** | **53.44** | **62.61** | **55.61** | **63.71** | **79.89** | **83.81** | 99.96 | 99.95 |

*Table 3.* **Cross-task adversarial transferability from image-text retrieval (ITR) to image captioning (IC).** Adversarial images are generated using CLIP$_{\text{ViT}}$ and evaluated on BLIP for captioning. **Lower values indicate stronger adversarial transferability** ($\downarrow$).

| Dataset | Method | BLEU-4 $\downarrow$ | METEOR $\downarrow$ | ROUGE_L $\downarrow$ | CIDEr $\downarrow$ | SPICE $\downarrow$ |
|---|---|---|---|---|---|---|
| Flickr30K | Clean | 18.28 | 21.50 | 37.65 | 43.90 | 15.25 |
| | SGA | 17.54 | 20.06 | 36.54 | 39.43 | 13.88 |
| | DRA | 17.24 | 19.93 | 36.48 | 39.40 | 13.75 |
| | SA-AET | 16.28 | 18.37 | 34.68 | 35.14 | 12.27 |
| | **DDGA** | **13.57** | **16.85** | **32.76** | **28.83** | **10.86** |
| MSCOCO | Clean | 32.20 | 30.00 | 49.50 | 100.30 | 22.80 |
| | SGA | 31.00 | 29.00 | 48.60 | 95.50 | 21.90 |
| | DRA | 30.60 | 28.60 | 48.20 | 93.70 | 21.60 |
| | SA-AET | 27.67 | 26.82 | 46.10 | 84.80 | 19.73 |
| | **DDGA** | **27.50** | **26.60** | **45.90** | **83.40** | **19.50** |

source VLM and testing them on unseen target models for both text-to-image (TR) and image-to-text (IR) retrieval on Flickr30K and MSCOCO. Tables 1 and 2 summarize the attack success rates (ASR) at the top-1 rank (R@1) of all methods. Overall, our method consistently achieves superior cross-model transferability across the majority of source–target pairs. On Flickr30K, DDGA yields clear improvements over prior AET-based approaches. For example, when transferring adversarial examples generated on CLIP$_{\text{ViT}}$ to ALBEF, DDGA improves ASR from **36.60%** to **41.19%** (TR) and from **50.44%** to **53.48%** (IR) compared with SA-AET. Similar gains are observed in other settings, such as CLIP$_{\text{CNN}}$ → TCL, where DDGA achieves **31.19% / 45.17%** (TR/IR), outperforming all baselines. The advantage of DDGA becomes more pronounced on the larger and more diverse MSCOCO dataset. Under the CLIP$_{\text{ViT}}$ → ALBEF setting, DDGA increases ASR from **57.64%** to **66.46%** (TR) and from **66.88%** to **72.53%** (IR), yielding substantial absolute gains of **8.82%** and **5.65%**,

respectively. Similar improvements are consistently observed across other transfer configurations, demonstrating that DDGA scales more effectively to complex data distributions. These results confirm that explicitly modeling and optimizing the perturbation distribution enables more reliable exploration of informative attack directions, leading to stronger and more stable adversarial transferability than AET-based methods. Additional experimental results can be found in Appendix C.

**Cross-Task Adversarial Transferability.** We further examine cross-task adversarial transferability by applying adversarial images generated for image–text retrieval (ITR) to image captioning (IC) task. Specifically, adversarial examples are crafted on CLIP$_{\text{ViT}}$ with the objective of attacking ITR, and are then directly evaluated on BLIP for caption generation without any task-specific adaptation. As shown in Table 3, our proposed DDGA consistently induces larger degradation in captioning performance than existing meth-

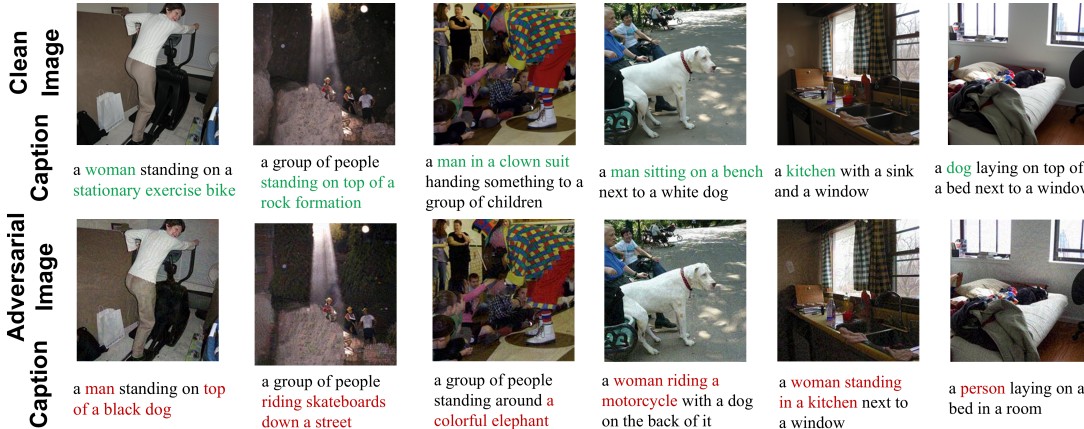

*Figure 3.* **Visualization of the image captioning task.** The first three example columns are from Flickr30K and the last three are from MSCOCO. Adversarial images are generated on both datasets using the CLIP$_{\text{ViT}}$ with the image-text retrieval task. We employ BLIP to both clean and adversarial images to produce captions for comparison. Green text indicates correct descriptions, while red text highlights errors induced by the attack.

ods on both Flickr30K and MSCOCO, as reflected by lower BLEU-4, METEOR, ROUGE_L, CIDEr, and SPICE scores. The performance drop is more pronounced on Flickr30K, while on the larger and more challenging MSCOCO benchmark, DDGA still maintains clear adversarial effectiveness despite relatively smaller margins over competing attacks. Overall, these results suggest that adversarial perturbations optimized for retrieval-oriented objectives can disrupt the semantic representations required by generation-based vision-language tasks, thereby demonstrating strong cross-task adversarial transferability.

**Qualitative Analysis of Cross-Task Adversarial Transferability.** Figure 3 provides qualitative comparisons between captions generated from clean images and adversarial images on the image captioning task. For clean inputs, BLIP generally produces accurate and semantically coherent captions that correctly capture the salient objects, attributes, and object relationships, such as "a person standing on a stationary exercise bike", "groups of people on a rock formation", or "a dog lying on a bed near a window". In contrast, when given adversarial images crafted for image-text retrieval, BLIP generates incomplete, misleading, or semantically inconsistent descriptions. The induced errors include misidentifying the primary subject (e.g., confusing a person with an animal), hallucinating nonexistent objects or entities, and misinterpreting the scene context by describing static scenes as dynamic activities. These qualitative results show that retrieval-oriented adversarial perturbations can disrupt the semantic grounding required for generation-based vision-language tasks, providing additional evidence of cross-task adversarial transferability beyond similarity-based retrieval.

**Adversarial Transferability in MLLMs.** Multimodal large language models (MLLMs) have demonstrated strong

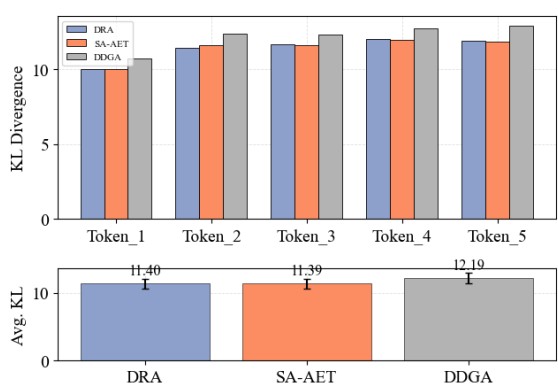

*Figure 4.* Transferability performance on Qwen3-VL-4B-Instruct.

joint vision-language reasoning capabilities, making their adversarial robustness an important concern. To evaluate whether adversarial perturbations crafted on contrastive VLMs can transfer to generative MLLMs, we apply adversarial images generated on CLIP$_{\text{ViT}}$ for image-text retrieval to a representative MLLM, *Qwen3-VL-4B-Instruct*. Following recent work, we quantify adversarial effectiveness by measuring the distributional shift of output tokens (Qi et al., 2025). Specifically, given the prompt *"Please describe the image, limited to 50 words"*, we feed both clean and adversarial images into the model and compute the KL divergence between their token distributions at each decoding step. Figure 4 reports the KL divergence of the top-5 output tokens as well as their average. As shown in Figure 4, the KL divergence generally increases with decoding depth, indicating that adversarial effects accumulate throughout the generation process. Moreover, DDGA yields consistently larger KL divergence than all baseline attacks across token positions and on average, suggesting stronger transferability from contrastive retrieval models to generative multimodal

*Table 4.* **Ablation study of DDGA on Flickr30K using the source model CLIP$_{ViT}$ for ITR task.** Policy learning captures optimal simplex mixing, covariance-guided perturbation enriches exploration, and text attack further amplifies cross-model transfer.

| Variant | ALBEF | | TCL | | CLIP$_{CNN}$ | |
|---|---|---|---|---|---|---|
| | TR R@1 | IR R@1 | TR R@1 | IR R@1 | TR R@1 | IR R@1 |
| DRA | 27.84 | 42.84 | 27.82 | 44.60 | 64.88 | 69.50 |
| SA-AET | 36.60 | 50.44 | 39.20 | 51.10 | **71.01** | 74.10 |
| DDGA w/o Policy (uniform $\alpha$) | 38.21 | 51.32 | 40.05 | 52.18 | 69.10 | 74.80 |
| DDGA w/o Covariance | 39.14 | 52.01 | 40.92 | 53.90 | 70.07 | 75.01 |
| DDGA w/o Text Attack | 36.77 | 51.90 | 39.21 | 50.05 | 66.82 | 72.10 |
| **DDGA (Full)** | **41.19** | **53.48** | **41.83** | **54.02** | 70.24 | **75.13** |

language models. Additional qualitative results on *Qwen3-VL-4B-Instruct* and *GPT-5.2* are provided in Appendix C, further supporting the generality of DDGA across different MLLMs.

## 5.2. Ablation Study

**Impact of Each Component.** Table 4 evaluates the contribution of each component in DDGA. Consistent with Theorem 3.2, replacing the learned Dirichlet policy with a uniform policy leads to a clear performance drop, indicating that effective gradient aggregation requires adaptive modeling of the perturbation distribution rather than predefined simplex sampling. Removing the covariance-guided perturbation further weakens transferability, suggesting that higher-order dispersion beyond the mean perturbation is important for exploring informative directions that may be overlooked by first-order aggregation. In addition, disabling adversarial text generation consistently reduces cross-model transferability, highlighting the necessity of jointly perturbing multimodal representations. When all components are enabled, DDGA achieves the best overall performance, validating that distributional policy learning and covariance-aware exploration jointly realize the theoretical advantages suggested by our second-order analysis.

**Impact of Injection Factor.** We analyze the impact of the covariance-guided perturbation injection factor $\lambda$ on adversarial transferability. As shown in Figure 5, when $\lambda$ is too small, the update direction is mainly dominated by the mean perturbation, limiting the benefit of exploration beyond the simplex expectation. Conversely, an excessively large $\lambda$ may overemphasize covariance-induced orthogonal perturbations, causing the update to deviate from the principal adversarial direction and making optimization less stable. A moderate value of $\lambda$ provides a better trade-off between these two effects, enabling complementary use of the expected gradient direction and covariance-driven exploration. Empirically, $\lambda = 0.05$ achieves the best overall performance on both TR and IR, and is therefore used as the default setting in all experiments.

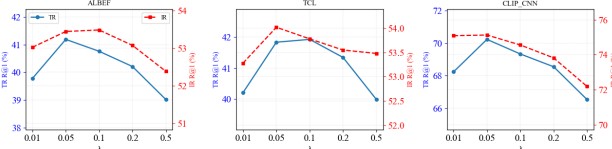

*Figure 5.* Sensitivity analysis of the covariance injection factor $\lambda$.

## 6. Conclusion

We propose DDGA, a distribution-aware adversarial attack framework that replaces heuristic sampling in AET-based methods with principled optimization over a learned perturbation distribution. By optimizing the expected adversarial objective and explicitly leveraging second-order structure to promote gradient diversity, DDGA generates adversarial examples with stronger transferability across models and tasks. Extensive experiments on image-text retrieval and image captioning demonstrate that DDGA achieves superior cross-model and cross-task transferability across diverse VLM architectures. These results highlight the importance of distributional modeling for adversarial transfer and suggest a promising direction for more reliable robustness evaluation of multimodal models. Despite its effectiveness, DDGA requires additional optimization over distribution parameters, resulting in moderate computational overhead compared with conventional attack methods. Extending distribution-aware optimization to broader multimodal perturbation spaces, including more tightly coupled image–text perturbations and generative MLLMs, remains an interesting direction for future work.

## Acknowledgements

This work was supported by National Natural Science Foundation of China (No.61702099), UIBE Excellent Young Scholar Project (21YQ10), and Scientific Research Laboratory of AI Technology and Applications, University of International Business and Economics.

## Impact Statement

This paper presents work whose goal is to advance the field of machine learning. There are many potential societal consequences of our work, none of which we feel must be specifically highlighted here.

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

# A. Proofs and Derivation.

### A.1. Proof of Lemma 3.1

*Proof.* For clarity, we focus on the image modality and consider the gradient of the adversarial loss with respect to the image perturbation. Let the loss be defined as

$$\ell_{\mathrm{CE}}(\mathbf{x} + \boldsymbol{\delta}) = -\log g_y(\mathbf{x} + \boldsymbol{\delta}), \tag{24}$$

where $\mathbf{x} \in \mathbb{R}^d$ denotes the clean image and $\boldsymbol{\delta}$ is adversarial perturbation. We denote the Jacobian and Hessian of $g_y$ with respect to the image input evaluated at $\mathbf{x}$ as

$$J_y(\mathbf{x}) \triangleq \nabla_{\mathbf{x}} g_y(\mathbf{x}) \in \mathbb{R}^d, \qquad H_y(\mathbf{x}) \triangleq \nabla_{\mathbf{x}}^2 g_y(\mathbf{x}) \in \mathbb{R}^{d \times d}. \tag{25}$$

Applying a second-order Taylor expansion of the scalar function $g_y(\mathbf{x} + \boldsymbol{\delta})$ around $\mathbf{x}$ yields

$$g_y(\mathbf{x} + \boldsymbol{\delta}) = g_y(\mathbf{x}) + J_y(\mathbf{x})^\top \boldsymbol{\delta} + \frac{1}{2}\boldsymbol{\delta}^\top H_y(\mathbf{x})\boldsymbol{\delta} + \mathcal{O}(\|\boldsymbol{\delta}\|^3). \tag{26}$$

Since adversarial perturbations are constrained within a small $\ell_\infty$-bounded neighborhood of the clean input, i.e., $\|\boldsymbol{\delta}\| \le \epsilon$, the third- and higher-order terms are negligible and can be safely omitted. Substituting the above expansion into $-\log g_y(\mathbf{x} + \boldsymbol{\delta})$ and expanding the logarithm up to second order gives

$$-\log g_y(\mathbf{x} + \boldsymbol{\delta}) \approx -\log g_y(\mathbf{x}) - \frac{J_y(\mathbf{x})^\top \boldsymbol{\delta}}{g_y(\mathbf{x})} - \frac{1}{2}\boldsymbol{\delta}^\top \left[ \frac{H_y(\mathbf{x})}{g_y(\mathbf{x})} - \frac{J_y(\mathbf{x})J_y(\mathbf{x})^\top}{g_y(\mathbf{x})^2} \right] \boldsymbol{\delta}. \tag{27}$$

Taking the gradient of the above expression with respect to $\boldsymbol{\delta}$ yields

$$\nabla_{\boldsymbol{\delta}} \ell_{\mathrm{CE}}(\mathbf{x} + \boldsymbol{\delta}) \approx -\frac{J_y(\mathbf{x})}{g_y(\mathbf{x})} - \left[ \frac{H_y(\mathbf{x})}{g_y(\mathbf{x})} - \frac{J_y(\mathbf{x})J_y(\mathbf{x})^\top}{g_y(\mathbf{x})^2} \right] \boldsymbol{\delta}, \tag{28}$$

which completes the proof. $\square$

### A.2. Proof of Theorem 3.2

*Proof.* Under the assumptions of Lemma 3.1, the gradient of the cross-entropy loss with respect to the perturbation admits the following second-order approximation:

$$\nabla_{\boldsymbol{\delta}} \ell_{\mathrm{CE}}(g(\mathbf{x} + \boldsymbol{\delta}), y) \approx \mathbf{a} + \mathbf{B}\boldsymbol{\delta}, \tag{29}$$

where

$$\mathbf{a} \triangleq -\frac{J_y(\mathbf{x})}{g_y(\mathbf{x})}, \qquad \mathbf{B} \triangleq -\left( \frac{H_y(\mathbf{x})}{g_y(\mathbf{x})} - \frac{J_y(\mathbf{x})J_y(\mathbf{x})^\top}{g_y(\mathbf{x})^2} \right). \tag{30}$$

Eq. (29) shows that, up to second order, the loss gradient is an affine function of the perturbation $\boldsymbol{\delta}$.

Taking expectation with respect to $\boldsymbol{\delta} \sim \Delta_{\mathbf{x}}$ and using the linearity of expectation, we obtain

$$\begin{aligned}
\mathbb{E}_{\boldsymbol{\delta} \sim \Delta_{\mathbf{x}}}\big[ \nabla_{\boldsymbol{\delta}} \ell_{\mathrm{CE}}(g(\mathbf{x} + \boldsymbol{\delta}), y) \big] &\approx \mathbf{a} + \mathbf{B}\mathbb{E}_{\boldsymbol{\delta} \sim \Delta_{\mathbf{x}}}[\boldsymbol{\delta}] \\
&= \mathbf{a} + \mathbf{B}\bar{\boldsymbol{\delta}},
\end{aligned} \tag{31}$$

where $\bar{\boldsymbol{\delta}}$ denotes the mean of the perturbation distribution.

On the other hand, evaluating the second-order approximation in Eq. (29) at $\boldsymbol{\delta} = \bar{\boldsymbol{\delta}}$ yields

$$\nabla_{\bar{\boldsymbol{\delta}}} \ell_{\mathrm{CE}}(g(\mathbf{x} + \bar{\boldsymbol{\delta}}), y) \approx \mathbf{a} + \mathbf{B}\bar{\boldsymbol{\delta}}. \tag{32}$$

Comparing the two expressions completes the proof. $\square$

### A.3. Proof of Theorem 3.4

*Proof.* Let $\mathbf{w} \sim \mathrm{Dir}(\boldsymbol{\alpha})$ be a Dirichlet random vector with parameters $\boldsymbol{\alpha} = (\alpha_1, \alpha_2, \alpha_3)$ and total concentration $S = \sum_{i=1}^{3} \alpha_i$. By definition, the Dirichlet-mixed input is given by

$$\mathbf{x}_{\mathrm{mix}} = \mathbf{V}\mathbf{w}, \tag{33}$$

where $\mathbf{V} = [\mathbf{x}, \tilde{\mathbf{x}}^{(t-1)}, \tilde{\mathbf{x}}^{(t)}] \in \mathbb{R}^{d \times 3}$.

**Expectation.** The expectation of $\mathbf{w}$ under the Dirichlet distribution is

$$\mathbb{E}_{\mathbf{w} \sim \mathrm{Dir}(\boldsymbol{\alpha})}[\mathbf{w}] = \frac{\boldsymbol{\alpha}}{S}. \tag{34}$$

Therefore, the mean of the mixed input is

$$\boldsymbol{\mu}_{\mathbf{x}} = \mathbb{E}_{\mathbf{w} \sim \mathrm{Dir}(\boldsymbol{\alpha})}[\mathbf{x}_{\mathrm{mix}}] = \mathbf{V}\,\mathbb{E}_{\mathbf{w} \sim \mathrm{Dir}(\boldsymbol{\alpha})}[\mathbf{w}] = \mathbf{V}\frac{\boldsymbol{\alpha}}{S}. \tag{35}$$

**Covariance.** By definition, the covariance of $\mathbf{x}_{\mathrm{mix}}$ is

$$\Sigma_{\mathbf{x}_{\mathrm{mix}}} = \mathbb{E}_{\mathbf{w} \sim \mathrm{Dir}(\boldsymbol{\alpha})}\big[(\mathbf{x}_{\mathrm{mix}} - \boldsymbol{\mu}_{\mathbf{x}})(\mathbf{x}_{\mathrm{mix}} - \boldsymbol{\mu}_{\mathbf{x}})^{\top}\big]. \tag{36}$$

Substituting $\mathbf{x}_{\mathrm{mix}} = \mathbf{V}\mathbf{w}$ and $\boldsymbol{\mu}_{\mathbf{x}} = \mathbf{V}\mathbb{E}_{\mathbf{w} \sim \mathrm{Dir}(\boldsymbol{\alpha})}[\mathbf{w}]$, we obtain

$$\Sigma_{\mathbf{x}_{\mathrm{mix}}} = \mathbb{E}_{\mathbf{w} \sim \mathrm{Dir}(\boldsymbol{\alpha})}\big[\mathbf{V}(\mathbf{w} - \mathbb{E}[\mathbf{w}])(\mathbf{w} - \mathbb{E}[\mathbf{w}])^{\top}\mathbf{V}^{\top}\big]. \tag{37}$$

Since $\mathbf{V}$ is deterministic, it can be factored out of the expectation, yielding

$$\Sigma_{\mathbf{x}_{\mathrm{mix}}} = \mathbf{V}\,\mathrm{Cov}(\mathbf{w})\,\mathbf{V}^{\top}. \tag{38}$$

The covariance matrix of a Dirichlet random vector $\mathbf{w} \sim \mathrm{Dir}(\boldsymbol{\alpha})$ is given by

$$C_{ij} = \mathrm{Cov}(w_i, w_j) = \begin{cases} \dfrac{\alpha_i(S - \alpha_i)}{S^2(S+1)}, & i = j, \\ -\dfrac{\alpha_i \alpha_j}{S^2(S+1)}, & i \neq j. \end{cases} \tag{39}$$

Denoting this covariance matrix by $\mathbf{C} \in \mathbb{R}^{3 \times 3}$ completes the proof. $\qquad\square$

### A.4. Policy Gradient Derivation

*Proof.* We derive the score-function gradient used in Eq. (14). Let $\mathbf{w} \sim \mathrm{Dir}(\boldsymbol{\alpha})$ with parameters $\boldsymbol{\alpha} = (\alpha_1, \dots, \alpha_K)$ and $S = \sum_{i=1}^{K} \alpha_i$. The Dirichlet density is given by

$$p(\mathbf{w} \mid \boldsymbol{\alpha}) = \frac{1}{B(\boldsymbol{\alpha})} \prod_{i=1}^{K} w_i^{\alpha_i - 1}, \tag{40}$$

where the normalization constant is

$$B(\boldsymbol{\alpha}) = \frac{\prod_{i=1}^{K} \Gamma(\alpha_i)}{\Gamma(S)}. \tag{41}$$

Taking the logarithm yields

$$\log p(\mathbf{w} \mid \boldsymbol{\alpha}) = \log \Gamma(S) - \sum_{i=1}^{K} \log \Gamma(\alpha_i) + \sum_{i=1}^{K} (\alpha_i - 1) \log w_i. \tag{42}$$

**Score Function.** Differentiating with respect to $\alpha_k$ gives

$$\frac{\partial}{\partial \alpha_k} \log p(\mathbf{w} \mid \boldsymbol{\alpha}) = \psi(S) - \psi(\alpha_k) + \log w_k, \tag{43}$$

where $\psi(\cdot)$ denotes the digamma function.

**Policy Gradient.** Using the score-function estimator, the gradient of the objective $J(\boldsymbol{\alpha}) = \mathbb{E}_{\mathbf{w}}[R(\mathbf{w})]$ is

$$\nabla_{\alpha_k} J(\boldsymbol{\alpha}) = \mathbb{E}_{\mathbf{w}} \left[ R(\mathbf{w}) \frac{\partial}{\partial \alpha_k} \log p(\mathbf{w} \mid \boldsymbol{\alpha}) \right]. \tag{44}$$

**Unbiased Estimator.** For a sampled $\mathbf{w}$, the resulting unbiased gradient estimator is

$$\widehat{\nabla_{\alpha_k} J} = R(\mathbf{w}) \left( \log w_k + \psi(S) - \psi(\alpha_k) \right), \tag{45}$$

which corresponds to Eq. (14) in the main text. $\square$

## B. Implementation Details

### B.1. Datasets and Models

**Dataset details.** Our experiments are conducted on two widely adopted multimodal benchmarks: Flickr30K (Plummer et al., 2015) and MSCOCO (Lin et al., 2014). Flickr30K consists of 31,783 images curated from the Flickr website, primarily depicting everyday activities and common scenes. MSCOCO (2014 version) is a more comprehensive dataset containing 123,287 images characterized by complex object-to-object interactions and diverse real-world contexts. For both datasets, each image is paired with five human-annotated captions, providing a rich linguistic corpus that describes the same visual content from multiple semantic perspectives.

To ensure consistency with prior work (Lu et al., 2023; Jia et al., 2025) on image-text retrieval and image captioning evaluation, we follow the standard Karpathy split protocol (Karpathy & Fei-Fei, 2015) for both benchmarks. Specifically, the Flickr30K dataset is partitioned into 29,000 training images, 1,000 validation images, and 1,000 test images. For MSCOCO, we follow the standard practice of incorporating the *restval* subset into the training set, resulting in a split of 113,287 images for training, 5,000 for validation, and 5,000 for testing. All the evaluation metrics, including retrieval Attack Success Rate (ASR) and captioning scores, are computed on the corresponding test sets.

**Model checkpoints.** We evaluate our proposed DDGA on a diverse set of VLMs, which we broadly categorize into aligned dual-encoders, fused multimodal encoders, and generative multimodal large language models (MLLMs). All the evaluations are conducted using official pre-trained weights to reflect practical evaluation settings.

*Source Model (Surrogate):* CLIP (Radford et al., 2021) serves as the primary surrogate model for generating adversarial examples. We utilize two architectural backbones: $\text{CLIP}_{\text{ViT}}$ (ViT-B/16 (Dosovitskiy et al., 2020)) and $\text{CLIP}_{\text{CNN}}$ (ResNet-101 (He et al., 2016)). This selection allows us to assess how perturbations transfer across different inductive biases—transformer-based global attention versus convolution-based local feature extraction.

*Target Models (Retrieval):* For image-text retrieval (ITR) evaluation, we utilize ALBEF (Li et al., 2021), TCL (Yang et al., 2022), and the surrogate backbones $\text{CLIP}_{\text{ViT}}$ and $\text{CLIP}_{\text{CNN}}$. Both ALBEF and TCL incorporate a 12-layer ViT-B/16 image encoder alongside two 6-layer Transformers dedicated to text and multimodal encoding. While sharing this architectural framework, TCL distinguishes itself by employing intra-modal self-supervision and local mutual information maximization objectives to capture more fine-grained structural information. We utilize official checkpoints fine-tuned on the Flickr30K and MSCOCO ITR task for all targets.

*Target Model (Captioning):* To assess cross-task transferability to generative tasks, we employ BLIP (Li et al., 2022). We utilize the $BLIP_{\text{ViT-B}}$ variant with the CapFilt-L configuration, using official weights fine-tuned on the MSCOCO image captioning dataset.

*Target Model (MLLM):* We further evaluate transferability to *Qwen3-VL-4B-Instruct* (Bai et al., 2025), an MLLM with 4.44 billion parameters. The model integrates a SigLIP2-Large vision encoder with a 36-layer LLM backbone.

*Table 5.* Additional cross-backbone transferability results on Flickr30K. We use CLIP$_{\text{ViT-L/14}}$ and ALBEF as source models and evaluate transferability across ALBEF, TCL, CLIP$_{\text{CNN}}$, CLIP$_{\text{ViT-B/16}}$, and CLIP$_{\text{ViT-L/14}}$. Higher R@1 indicates stronger adversarial effectiveness and transferability.

| Source | Attack | ALBEF | | TCL | | CLIP$_{\text{CNN}}$ | | CLIP$_{\text{ViT-B/16}}$ | | CLIP$_{\text{ViT-L/14}}$ | |
|---|---|---|---|---|---|---|---|---|---|---|---|
| | | TR | IR | TR | IR | TR | IR | TR | IR | TR | IR |
| CLIP$_{\text{ViT-L/14}}$ | DRA | 23.98 | 38.45 | 25.92 | 39.74 | 53.38 | 60.62 | 60.12 | 67.46 | 100.00 | 99.93 |
| | SA-AET | 30.34 | 44.13 | 31.19 | 44.17 | 55.43 | 63.02 | 66.63 | 73.00 | 100.00 | 99.57 |
| | **DDGA** | **37.85** | **51.87** | **39.62** | **52.24** | **61.31** | **69.16** | **72.15** | **77.60** | 99.76 | 96.64 |
| ALBEF | DRA | 99.90 | 99.98 | 91.57 | 91.17 | 49.55 | 59.01 | 46.26 | 56.80 | 30.66 | 49.85 |
| | SA-AET | 99.90 | 99.95 | 96.42 | 96.02 | 57.22 | 65.59 | 55.58 | 63.89 | 34.75 | 53.52 |
| | **DDGA** | **99.90** | 99.95 | **97.05** | **97.60** | **68.45** | **73.89** | **67.73** | **72.97** | **44.28** | **64.93** |

*Table 6.* Cross-model adversarial transfer success rate (%) under different image perturbation budgets $\epsilon_I$ on the Flickr30K ITR task. Adversarial examples are generated on CLIP$_{\text{ViT}}$ and evaluated on ALBEF, TCL, and CLIP$_{\text{CNN}}$ for both text-to-image (TR) and image-to-text (IR) retrieval.

| Source | $\epsilon_I$ | ALBEF | | TCL | | CLIP$_{\text{CNN}}$ | |
|---|---|---|---|---|---|---|---|
| | | TR R@1 | IR R@1 | TR R@1 | IR R@1 | TR R@1 | IR R@1 |
| CLIP$_{\text{ViT}}$ | 4/255 | 38.98 | 49.88 | 39.77 | 51.28 | 68.04 | 72.09 |
| | 6/255 | 40.07 | 52.39 | 40.42 | 53.39 | 69.55 | 73.28 |
| | 8/255 | 41.19 | 53.48 | 41.83 | 54.02 | 70.24 | 75.13 |
| | 10/255 | 43.39 | 55.27 | 44.75 | 55.89 | 72.39 | 77.81 |
| | 12/255 | 44.78 | 56.45 | 45.09 | 56.74 | 75.03 | 79.54 |

**Additional evaluation details.** For image-text retrieval (ITR), the Attack Success Rate (ASR) is calculated over the full test sets of Flickr30K and MSCOCO. The ASR at Top-1 (R@1) is defined as the percentage of images for which the correctly retrieved caption in a clean setting is no longer retrieved at the first rank after the adversarial perturbation is applied. The retrieval performance is measured by computing the similarity scores between each adversarial image and all possible text candidates in the test set. For fusion models like ALBEF and TCL, this involves a two-stage process: An initial ranking based on unimodal contrastive scores followed by a re-ranking stage using the multimodal encoder.

For image captioning (IC), the evaluation transitions from a discriminative task to a generative one. Adversarial images generated for retrieval are provided to the BLIP model to generate captions. The quality of these captions is measured using BLEU-4(Papineni et al., 2002), METEOR(Banerjee & Lavie, 2005), ROUGE_L(Lin, 2004), CIDEr(Vedantam et al., 2015), and SPICE(Anderson et al., 2016).

## C. Additional Experiments Results

### C.1. Additional Cross-Backbone Transferability

We further evaluate cross-backbone adversarial transferability on Flickr30K by using CLIP$_{\text{ViT-L/14}}$ and ALBEF as additional source models. This setting complements the main experiments by introducing a larger ViT backbone and a fused vision-language model as surrogates, and evaluates adversarial transfer to ALBEF, TCL, CLIP$_{\text{CNN}}$, CLIP$_{\text{ViT-B/16}}$, and CLIP$_{\text{ViT-L/14}}$. Table 5 shows that DDGA maintains its advantage under more source-target settings. When CLIP$_{\text{ViT-L/14}}$ is used as the source model, DDGA improves transfer to fused VLMs over SA-AET by **+7.51/+7.74** points on ALBEF and **+8.43/+8.07** points on TCL for TR/IR, respectively. The improvement is more pronounced when ALBEF is used as the source model and the target is changed to CLIP variants with different visual encoders: DDGA improves ALBEF → CLIP$_{\text{CNN}}$ by **+11.23/+8.30** points and ALBEF → CLIP$_{\text{ViT-L/14}}$ by **+9.53/+11.41** points for TR/IR. These results indicate that DDGA remains effective across different backbone scales, encoder architectures, and vision-language training paradigms, rather than relying on backbone overlap between the surrogate and target models.

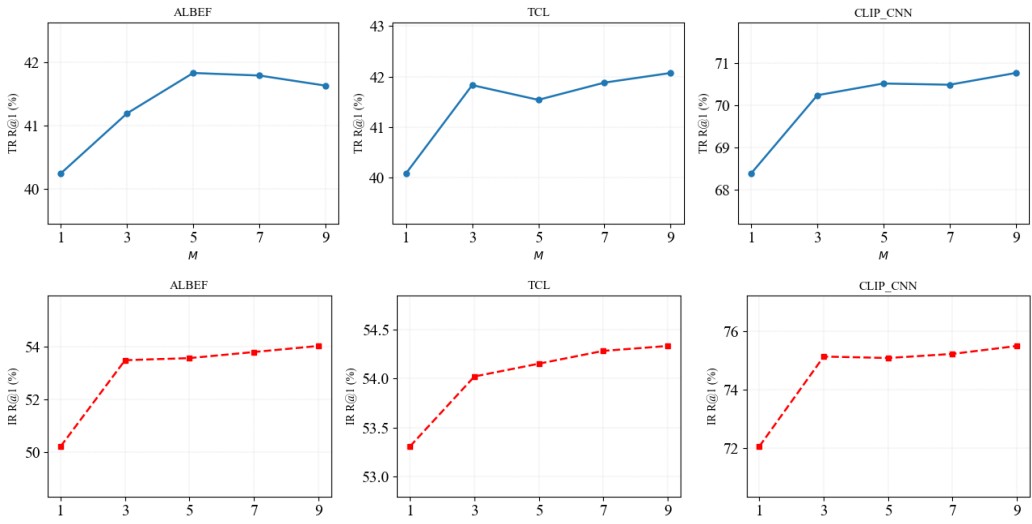

*Figure 6.* Effect of the number of mixed adversarial candidates $M$ on transfer attack success rate (R@1) for the image-text retrieval (ITR) task on the Flickr30K dataset. The top row reports text-to-image retrieval (TR), while the bottom row reports image-to-text retrieval (IR). Results are shown for transfers from CLIP$_{\text{ViT}}$ to ALBEF, TCL, and CLIP$_{\text{CNN}}$.

## C.2. Adversarial Transferability *w.r.t.* Diverse Perturbation Budgets.

Table 6 presents the adversarial transferability of DDGA under varying image perturbation budgets $\epsilon_I$, where attacks are generated on CLIP$_{\text{ViT}}$ and evaluated across multiple target VLMs. Across all models and retrieval settings, DDGA exhibits a consistent and monotonic increase in attack success rate as $\epsilon_I$ grows, indicating stable and effective utilization of larger perturbation budgets. Importantly, the performance gains are smooth and well-behaved, without abrupt saturation or instability, suggesting that DDGA exploits additional perturbation capacity in a structured manner rather than relying on fragile directions. The improvement is especially notable in cross-architecture transfer settings such as CLIP$_{\text{ViT}}$ → ALBEF and TCL, highlighting the model-agnostic nature of the learned adversarial perturbations. Transferability to CLIP$_{\text{CNN}}$ also increases steadily, demonstrating robustness across architectural variations. Moreover, DDGA maintains comparable relative gains for both text-to-image and image-to-text retrieval tasks across all budgets, indicating that the learned perturbation distribution disrupts cross-modal alignment in a balanced and task-agnostic manner. Overall, these results confirm that DDGA scales reliably with perturbation radius and consistently enhances adversarial transferability across models and tasks.

## C.3. Adversarial Transferability *w.r.t.* Mixed Adversarial Candidates $M$.

We investigate the impact of the number of mixed adversarial candidates $M$ on adversarial transferability, where $M$ determines the size of the simplex used for distributional aggregation in DDGA. Figure 6 reports the transfer attack success rate (R@1) under varying $M$ when transferring adversarial examples from CLIP$_{\text{ViT}}$ to ALBEF, TCL, and CLIP$_{\text{CNN}}$ on Flickr30K ITR task. As $M$ increases, DDGA consistently achieves higher attack success rates for both text-to-image (TR) and image-to-text (IR) retrieval across all target models. This trend indicates that incorporating more historical adversarial candidates enriches the simplex structure and allows the learned Dirichlet distribution to better capture transferable perturbation directions. However, larger values of $M$ also incur increased computational cost, as each additional candidate requires extra gradient evaluations and distributional updates during optimization. Empirically, we observe that the performance gains diminish when $M$ exceeds 3, while the computational overhead continues to grow. Taking both adversarial effectiveness and efficiency into account, we adopt $M = 3$ in all experiments, which provides a favorable balance between transferability performance and computational cost.

## C.4. Adversarial Transferability against Robustly Defended VLMs.

Table 7 reports the transfer attack performance on TeCoA, a CLIP-based model explicitly fine-tuned with adversarial examples for robustness. Compared with standard transfer baselines, TeCoA significantly suppresses the effectiveness of prior attacks, highlighting the challenge of attacking defended VLMs. Despite this strengthened defense, DDGA consistently

*Table 7.* Performance on Robust CLIP model TeCoA using the Flickr30k dataset. The source model is CLIP_ViT, and the target model is TeCoA, a defense model fine-tuned with adversarial examples.

| Method | TR R@1 | TR R@5 | TR R@10 | IR R@1 | IR R@5 | IR R@10 |
|--------|--------|--------|---------|--------|--------|---------|
| SGA | 51.84 | 24.44 | 17.68 | 55.07 | 37.45 | 28.97 |
| DRA | 59.59 | 34.59 | 26.75 | 64.35 | 47.26 | 39.04 |
| SA-AET | 68.98 | 43.98 | 36.85 | 72.55 | 55.62 | 48.15 |
| **DDGA** | **75.51** | **55.83** | **48.74** | **78.33** | **64.22** | **57.14** |

*Table 8.* Additional semantic evaluation on Qwen3-VL-4B-Instruct. Qwen3-Reranker-4B measures retrieval similarity between the original text and the generated output, where lower scores indicate stronger semantic disruption. GLM-4.7-Flash is used as an LLM judge to rank semantic relevance, where higher ranks indicate lower relevance.

| Method | Qwen3-Reranker-4B $\downarrow$ | GLM-4.7-Flash Rank $\uparrow$ |
|--------|-------------------------------|-------------------------------|
| DRA | $0.22 \pm 0.37$ | $1.89 \pm 0.75$ |
| SA-AET | $0.17 \pm 0.34$ | $1.95 \pm 0.87$ |
| **DDGA** | $\mathbf{0.15 \pm 0.32}$ | $\mathbf{2.16 \pm 0.80}$ |

achieves the highest attack success rates across all retrieval metrics in both text-to-image (TR) and image-to-text (IR) settings. In particular, DDGA improves TR R@1 by a large margin over SA-AET (75.51% vs. 68.98%) and yields similarly substantial gains on IR R@1 (78.33% vs. 72.55%). The performance gap further widens for higher recall thresholds (R@5 and R@10), indicating that DDGA not only disrupts the top-ranked retrieval but also degrades the overall retrieval structure more effectively. These results suggest that distribution-aware gradient aggregation produces adversarial perturbations that generalize beyond standard CLIP models and remain effective even against adversarially trained defenses. In contrast to sampling-based AET variants, DDGA learns more stable and transferable perturbation directions, which appear harder for defense mechanisms such as TeCoA to neutralize.

### C.5. Additional Semantic Evaluation on MLLMs

We further provide semantic-level evaluation on Qwen3-VL-4B-Instruct to complement the token-distribution KL divergence analysis in the main paper. Specifically, we use Qwen3-Reranker-4B to measure the retrieval similarity between the original text and the generated output, where lower similarity indicates stronger semantic deviation from the original image-text pair. We also employ GLM-4.7-Flash as an LLM judge to rank the semantic relevance of generated outputs, where a higher rank indicates lower semantic relevance. As shown in Table 8, DDGA produces the lowest retrieval similarity and the highest semantic irrelevance rank among all methods. Compared with the strongest baseline SA-AET, DDGA further reduces the Qwen3-Reranker-4B similarity from $0.17 \pm 0.34$ to $\mathbf{0.15 \pm 0.32}$ and increases the GLM-4.7-Flash rank from $1.95 \pm 0.87$ to $\mathbf{2.16 \pm 0.80}$. The larger gap against DRA on retrieval similarity, from $0.22 \pm 0.37$ to $\mathbf{0.15 \pm 0.32}$, further suggests that DDGA induces stronger semantic drift in MLLM generation. These results show that the attack effect is not limited to token-level distribution shift, but also manifests as semantic degradation.

### C.6. Qualitative Examples on MLLM

Figures 7 and 8 show additional qualitative adversarial examples on two representative MLLMs, including the open-source *Qwen3-VL-4B-Instruct* and the closed-source *GPT-5.2*. The adversarial images are generated on CLIP_ViT for image–text retrieval and directly transferred to MLLMs for visual instruction following, without any model-specific optimization. As shown in the figures, DDGA induces clear semantic shifts in the generated descriptions, leading to errors such as misidentifying the main subject, hallucinating nonexistent objects, and misinterpreting scene context or object relations. Although the generated responses remain linguistically fluent, they become semantically inconsistent with the visual content, indicating that the perturbations can disrupt high-level visual grounding rather than only low-level perception. These qualitative results, observed on both open-source and closed-source MLLMs, further support the generality of DDGA and complement the quantitative KL-divergence analysis in the main paper.

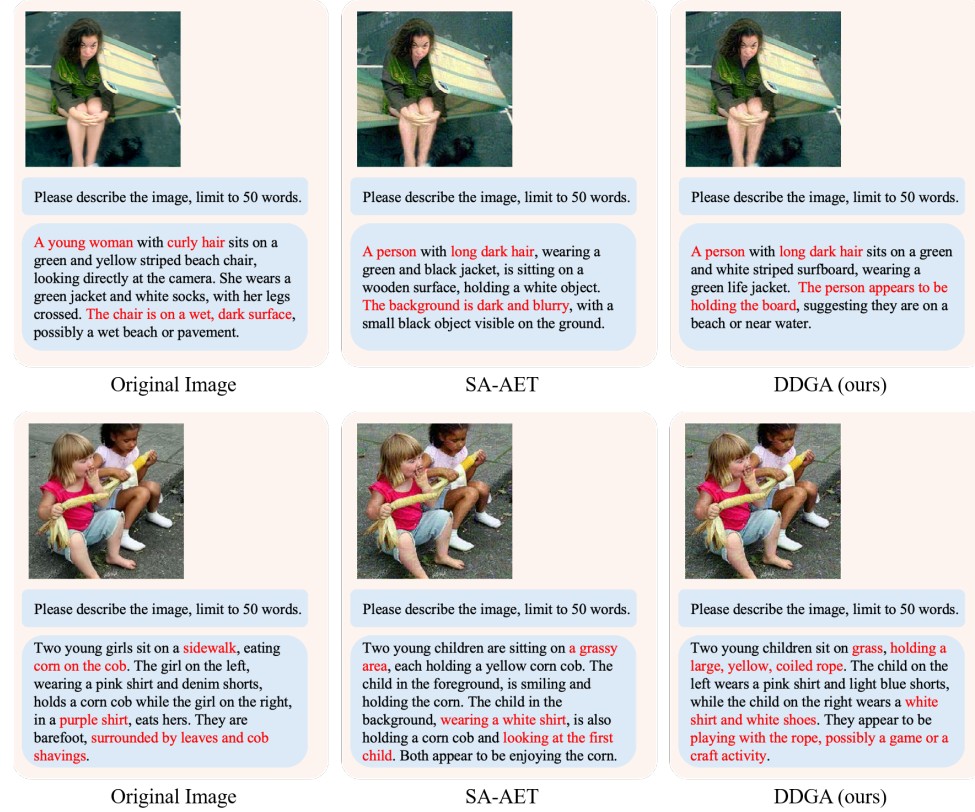

*Figure 7.* Qualitative adversarial examples on Qwen3-VL-4B-Instruct.

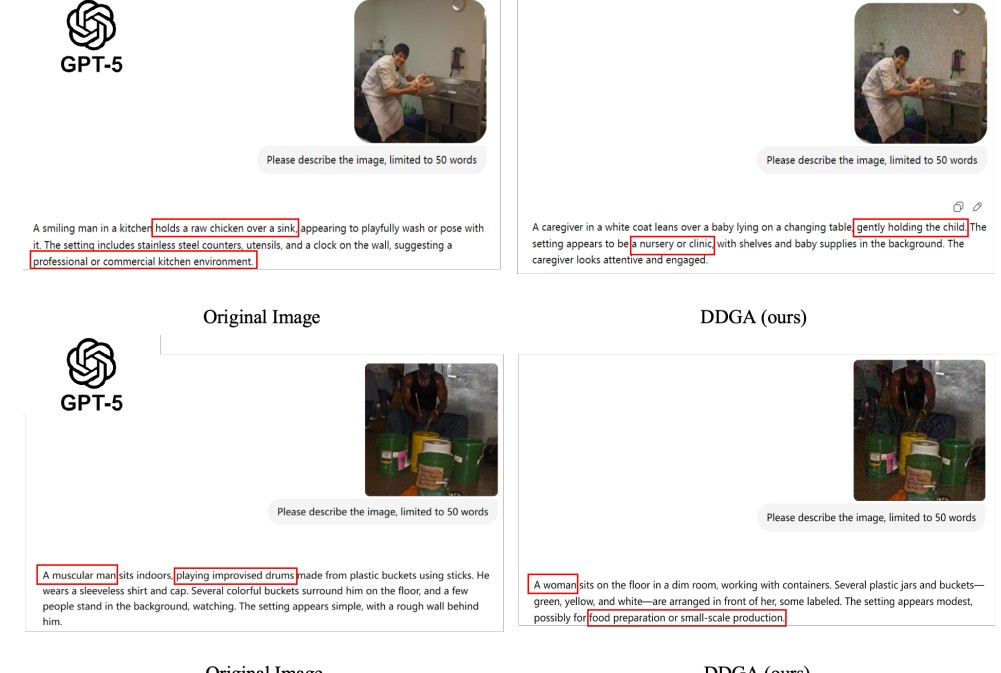

*Figure 8.* Qualitative adversarial examples on GPT-5.2.

