# OpenReview forum: "DDGA: Dirichlet Distributional Gradient Aggregation for Transferable Vision-Language Adversarial Attacks"
_ICML.cc/2026/Conference — ICML 2026 regular_

### Official Review · Reviewer_sXwC · 2026-03-02

**Soundness:** 3
**Presentation:** 3
**Significance:** 3
**Originality:** 3
**Overall Recommendation:** 4
**Confidence:** 3

**Summary:**

This paper studies transferable multimodal adversarial attacks for vision–language models and proposes Dirichlet Distributional Gradient Aggregation (DDGA). The method learns a Dirichlet distribution to aggregate gradients and generate diverse perturbations, aiming to improve attack transferability. Experiments on Flickr30K and MSCOCO show improved cross-model and cross-task attack performance over prior AET-based methods.

**Compliance With Llm Reviewing Policy:**

Affirmed.

**Final Justification:**

My concerns have been satisfactorily resolved in the authors' rebuttal. Considering that adversarial attack and defense remain an under-explored area compared to traditional deep learning settings, I maintain my score of weak acceptance to encourage this important line of research, which I believe deserves more attention.

**Key Questions For Authors:**

See weaknesses.

**Limitations:**

The paper would benefit from an analysis of the computational cost of the proposed method. In particular, a discussion of time and space complexity would help assess its practicality.

**Strengths And Weaknesses:**

Strengths
- Proposes a principled reformulation of AET sampling by learning the perturbation distribution instead of relying on heuristic gradient aggregation.
- Introduces a Dirichlet policy with covariance-guided exploration to improve gradient diversity and attack transferability.
- Demonstrates consistent improvements across multiple VLM architectures and multimodal tasks.

Weaknesses
1. The meaning of the differently colored sample images in Figure 1 is unclear. In particular, it is not well explained what distinguishes the two types of the sample image, nor what the variation in background shading in the right subfigure represents.
2. 'DRA' introduced in Section 2.2 is not explicitly defined, which may affect readability.
3. The paper would benefit from an analysis of the computational cost of the proposed method. In particular, a discussion of time and space complexity would help assess its practicality.

---

> ### Author Rebuttal · Authors · 2026-03-29
>
> **W1**: We sincerely thank the reviewer for pointing out the lack of clarity in Figure 1, and agree that the current visualization can be improved for better interpretability.
>
> In the left subfigure (DRA), the sample images in different colors represent *all the samples sampled from the simplex formed by the historical adversarial samples*, while the brown sample image denotes the final selected optimal one. On this basis, the samples in pink represent those augmented using the brown optimal sample. In the right subfigure (DDGA), the same set of samples is instead associated with a learned Dirichlet distribution over simplex weights. The variation in background shading visualizes the probability density over the simplex, where darker regions indicate higher probability mass. This illustrates how DDGA adaptively concentrates on more informative mixing directions, rather than relying on uniform or random sampling as in DRA. Based on the learned Dirichlet distribution, *we directly employ the expectation of the distribution as the optimal sample image, and perform the augmentation operation on this basis.* Moreover, Figure 2 further illustrates how the learned Dirichlet distribution evolves during optimization and guides the aggregation process, which complements the conceptual comparison shown in Figure 1. We believe that **viewing both figures together** provides a clearer understanding of how DDGA transitions from heuristic sampling to distribution-aware optimization.
>
> We will revise the figures by adding explicit legends, annotations, and a more detailed caption to clearly explain the meaning of colors and shading in final version. If any ambiguity remains, we would be happy to further clarify this in the discussion phase.
>
> **W2**: We totally agree that the current presentation of DRA in Section 2.2 could be improved for clarity. We will carefully revise the Background section to provide a clearer and more explicit description of DRA, including its key components and role within AET-style methods, in order to enhance readability. In addition, we will include the detailed pseudo-code of DRA in the supplementary material in the revised version, so that readers can more easily follow the algorithm.
>
> **W3**: We agree that a clear analysis for computational cost is essential. For complexity, we denote $T$ as number of attack iterations, $V$ as number of augmentation views, $K$ as number of sampled points, $D$ denote image dimension, and $d$ denote Dirichlet dimension. Let $C_{f}$ and $C_{b}$ as forward and backword costs, and $C_{g}=C_{f}+C_{b}$ as gradient costs. For strong baseline DRA and SA-AET, each iteration requires $KV$ gradient evaluations, leading to complexity ${\cal{O}}(TKVC_{g})$. In contrast, DDGA performs only one gradient evaluation per iteration, and replaces the remaining operations with forward-only evaluations and lightweight distribution updates, resulting in complexity: ${\cal{O}}(T(KVC_{f}+C_{g}+d+KD))$, where $d$ and $KD$ correspond to lightweight policy update and calculation of covariance-guided matrix. Since $C_{g}>C_{f}$, DDGA substantially reduces the number of expensive gradient computations for image-side attack.
>
> We also measure the empirical runtime and the average performance of each method with source model CLIP_ViT on both Flickr30K and MSCOCO:
>
> |Method|Time (s/epoch)|Complexity|Flickr30K|MSCOCO|
> |-|-|-|-|-|
> |SGA|4.799|${\cal{O}}(TVC_{g})$|38.82 $\pm$ 15.40|56.02 $\pm$ 13.71|
> |DRA|5.304|${\cal{O}}(TKVC_{g})$|46.25 $\pm$ 17.78|65.24 $\pm$ 13.76|
> |SA-AET|7.314|${\cal{O}}(TKVC_{g})$|52.74 $\pm$ 15.72|69.61 $\pm$ 12.82|
> |DDGA (ours)|8.273|${\cal{O}}(T(KVC_{f}+C_{g}+d+KD))$|55.98 $\pm$ 14.14 (3.24 $\uparrow$)|75.13 $\pm$ 10.71 (5.52 $\uparrow$)|
>
> Although DDGA reduces the number of expensive gradient computations compared to prior methods, it introduces additional forward passes and lightweight distribution updates (e.g., Dirichlet sampling and aggregation), which incur extra computational overhead in practice. Moreover, these operations are less optimized than standard backward passes on modern hardware, leading to slightly higher wall-clock time. In practice, although DDGA introduces modest overhead (8.273s vs 7.314s), it achieves substantially better performance (**+3.24%** on Flickr30K and **+5.52%** on MSCOCO over SA-AET), demonstrating a favorable efficiency–performance trade-off. In addition, DDGA exhibits lower variance compared to prior methods, indicating more stable transferability across different target models. This suggests that the learned distributional policy not only improves attack strength but also enhances robustness and consistency.
>
> Moreover, we present qualitative visualizations from Qwen3-VL-4B-Instruct, LLaVA-v1.6-Mistral-7B, and GPT-5.2 in Figure 1-3 in https://anonymous.4open.science/r/icml8075-70DC/README.md, which offer more intuitive evidence of generation quality degradation. We kindly encourage the reviewer to take a look for additional insights.

---

> > ### Author Rebuttal · Reviewer_sXwC · 2026-04-01
> >
> > Thank you for the authors' detailed response. My concerns have been adequately addressed.

---

> > > ### Author Response · Authors · 2026-04-06
> > >
> > > We sincerely thank you for the recognition of our work and your constructive feedback.
> > > We will carefully revise our manuscript based on the suggestions provided to further improve its quality and clarity.
> > > We greatly appreciate your time and effort.

---

### Official Review · Reviewer_qr7b · 2026-03-04

**Soundness:** 4
**Presentation:** 4
**Significance:** 3
**Originality:** 3
**Overall Recommendation:** 4
**Confidence:** 3

**Summary:**

The paper addresses adversarial examples on VLMs for multimodal tasks. Previous SOTA builds on AET, e.g., DRA, and uses finite examples (current clean image and historical adversarial images) to create adversarial samples. The authors show in Theorem 3.2 that averaging gradients is approximately equivalent to evaluating a single gradient at the mean of the perturbation distribution instead of individual samples. Motivated by this, the authors propose to parameterize the simplex mixing weights with a Dirichlet distribution and optimize its parameters via policy gradient using CE loss as the reward. The proposed method, abbreviated as DDGA, is evaluated on MSCOCO and Flickr30K for image-text retrieval and image captioning, showing improvements over previous SOTA.

**Compliance With Llm Reviewing Policy:**

Affirmed.

**Final Justification:**

Thank you for the rebuttal, I believe that most of my concerns are resolved.

**Key Questions For Authors:**

1. Does the method perform similarly at other epsilon values? Can you measure the approximation error between the expected gradient and the gradient at the mean across different perturbation magnitudes?

2. Can you evaluate the method on other models, e.g., ALBEF, BLIP, SigLIP, EVA-CLIP or larger backbones (ViT-L) as source models, to further validate the generality of the approach?

3. Can you provide a wall-clock time comparison against the baselines (DRA, SA-AET vs DDGA) to clarify the computational overhead introduced by the Dirichlet policy optimization?

4. Can you add evaluation against existing VLM adversarial defenses to assess the robustness of the generated adversarial examples?

**Limitations:**

The authors acknowledge the computational overhead and the restriction to image-modality distribution optimization in the conclusion. The societal impact statement is generic.

**Strengths And Weaknesses:**

The paper is well-written and technically sound. It provides a clear motivation along with the limitations of current SOTA, which relies on finite samples to create adversarial samples. Approximating the distribution instead is intuitive and well-explained. The connection to theory is clean. The authors evaluate their method on a variety of benchmarks and have notable gains over previous SOTA.

My only concern is focused around the experimental setup used to evaluate the method. It is currently based on CLIP ViT-B and RN-101 to create adversarial samples, and the testbed is somewhat limited. Models like ALBEF are not tested. Also, no statistical significance (mean, std) is reported over multiple runs.

---

> ### Author Rebuttal · Authors · 2026-03-30
>
> **Q1**: We respectfully clarify that we already report attack transferability of our method under different $\epsilon$ values on Appendix C.2. We present the average ASR of both TR and IR with source model CLIP_ViT on Flickr30K (the full version can be found in Appendix Table 5):
>
> |$\epsilon$/255|ALBEF|TCL|CLIP_CNN|
> |-|-|-|-|
> |4|44.43|45.53|70.07|
> |6|46.23|46.91|71.42|
> |8|47.34|47.93|72.69|
> |10|49.33|50.32|75.10|
> |12|50.62|50.92|77.29|
>
> We observe that DDGA consistently maintains strong performance across a range of $\epsilon$. We will highlight this experiment more prominently for better readability.
>
> From theoretical perspective, as perturbation magnitude increases, the approximation error introduced by second-order expansion also grows (e.g., scaling with higher-order terms), which may lead to slight degradation or marginal variation in performance at larger $\epsilon$. Specifically, under $\ell_{\infty}$ constraint, there exists a constant $C>0$ (independent of $\epsilon$, determined only by third-order smoothness of $g_y$ at $x$) such that the approximation error satisfies $||{\varepsilon}({\delta})||_\infty \leq C \epsilon^2$.
>
> **Q2**: To further validate generality, we choose CLIP_ViT-L/14 and ALBEF as source model and evaluate cross-backbone transfer in both directions (e.g., CLIP_ViT-L→ALBEF and ALBEF→CLIP variants) on Flickr30K:
>
> ||Target|ALBEF||TCL||CLIP_CNN||CLIP_ViT-B/16||CLIP_ViT-L/14||
> |-|-|:-|:-|:-|:-|:-|:-|:-|:-|:-|:-|
> |Source|Method|TR|IR|TR|IR|TR|IR|TR|IR|TR|IR|
> |CLIP_ViT-L/14|DRA|23.98|38.45|25.92|39.74|53.38|60.62|60.12|67.46|100.00|99.93|
> ||SA-AET|30.34|44.13|31.19|44.17|55.43|63.02|66.63|73.00|100.00|99.57|
> ||DDGA (ours)|37.85|51.87|39.62|52.24|61.31|69.16|72.15|77.60|99.76|96.64|
> |ALBEF|DRA|99.90|99.98|91.57|91.17|49.55|59.01|46.26|56.80|30.66|49.85|
> ||SA-AET|99.90|99.95|96.42|96.02|57.22|65.59|55.58|63.89|34.75|53.52|
> ||DDGA (ours)|99.90|99.95|97.05|97.60|68.45|73.89|67.73|72.97|44.28|64.93|
>
> As shown, DDGA outperforms DRA and SA-AET under both settings, and gains remain significant even when transferring between heterogeneous backbones, indicating that improvements are not tied to specific architectures. These results show that DDGA generalizes well across different model families and scales.
>
> More qualitative visualizations from MLLM Qwen3-VL-4B-Instruct, LLaVA-v1.6-Mistral-7B, and GPT-5.2 are presented in Figure 1-3 in https://anonymous.4open.science/r/icml8075-70DC/README.md, which offer more evidence of generality of DDGA.
>
> **Q3**: For complexity, we denote $T$ as number of attack iterations, $V$ as number of augmentation views, $K$ as number of sampled points, $D$ denote image dimension, and $d$ denote Dirichlet dimension. Let $C_{f}$ and $C_{b}$ as forward and backword costs, and $C_{g}=C_{f}+C_{b}$ as gradient costs. For strong baseline DRA and SA-AET, each iteration requires $KV$ gradient evaluations, leading to complexity ${\cal{O}}(TKVC_{g})$. In contrast, DDGA performs only one gradient evaluation per iteration, and replaces remaining operations with forward-only evaluations and lightweight distribution updates, resulting in complexity: ${\cal{O}}(T(KVC_{f}+C_{g}+d+KD))$, where $d$ and $KD$ correspond to lightweight policy update and calculation of covariance-guided matrix. Since $C_{g}>C_{f}$, DDGA substantially reduces the number of expensive gradient computations for image-side attack.
>
> We also measure empirical runtime and the average performance of each method with source model CLIP_ViT on both Flickr30K and MSCOCO:
>
> |Method|Time (s/epoch)|Complexity|Flickr30K|MSCOCO|
> |-|-|-|-|-|
> |SGA|4.799|${\cal{O}}(TVC_{g})$|38.82 $\pm$ 15.40|56.02 $\pm$ 13.71|
> |DRA|5.304|${\cal{O}}(TKVC_{g})$|46.25 $\pm$ 17.78|65.24 $\pm$ 13.76|
> |SA-AET|7.314|${\cal{O}}(TKVC_{g})$|52.74 $\pm$ 15.72|69.61 $\pm$ 12.82|
> |DDGA (ours)|8.273|${\cal{O}}(T(KVC_{f}+C_{g}+d+KD))$|55.98 $\pm$ 14.14 (3.24 $\uparrow$)|75.13 $\pm$ 10.71 (5.52 $\uparrow$)|
>
> Although DDGA reduces the number of expensive gradient computations, it introduces additional forward and lightweight distribution operations, leading to slightly higher wall-clock time (see column 2). In practice, although DDGA introduces modest overhead (8.273s vs 7.314s), it achieves better performance (**+3.24%** on Flickr30K and **+5.52%** on MSCOCO over SA-AET), showing a favorable efficiency–performance trade-off.
>
> **Q4**: We respectfully note that we already evaluated DDGA against an adversarial defense model, TeCoA, a CLIP-based model explicitly fine-tuned for robustness using adversarial examples (fully version can be found in Section C.4 and Table 6 in Appendix):
>
> |Method|TR R@1|IR R@1|
> |-|-|-|
> |SGA|51.84|55.07|
> |DRA|59.59|64.35|
> |SA-AET|68.98|72.55|
> |DDGA (ours)|75.51|78.33|
>
> The results show that DDGA achieves higher ASR than baselines under both IR and TR tasks, which indicates that adversarial examples generated by DDGA remain effective even against robust models. We further highlight this experiment more prominently in the revised version.

---

> > ### Author Rebuttal · Reviewer_qr7b · 2026-04-01
> >
> > Thank you for the response. My concerns have been adequately addressed.

---

> > > ### Author Response · Authors · 2026-04-06
> > >
> > > We sincerely thank you for the recognition of our work and your constructive feedback.
> > > We will carefully revise our manuscript based on the suggestions provided to further improve its quality and clarity.
> > > We greatly appreciate your time and effort.

---

### Official Review · Reviewer_ascp · 2026-03-06

**Soundness:** 3
**Presentation:** 4
**Significance:** 3
**Originality:** 3
**Overall Recommendation:** 4
**Confidence:** 4

**Summary:**

The paper studies transferable adversarial attacks on vision-language models. The main idea is to replace heuristic simplex sampling in AET-style attacks with an explicitly learned perturbation distribution. Concretely, the method parameterizes simplex mixing weights with a Dirichlet policy, optimizes the policy with policy gradient, and further uses the covariance of the learned distribution to add an orthogonal exploration term. Experiments on retrieval, captioning, and transfer to a defended model show consistent gains over DRA and SA-AET.

**Compliance With Llm Reviewing Policy:**

Affirmed.

**Key Questions For Authors:**

1. Since the text attack is reused from prior work, how much of the final gain really comes from the proposed image-side distribution learning?

2. How sensitive is the method to the policy optimization itself? For example, does the learned Dirichlet policy converge stably across different samples, or does it have high variance?

**Limitations:**

Limitations and potential negative impacts have been sufficiently discussed in the paper.

**Strengths And Weaknesses:**

**Strengths**

1. The paper has a clear motivation. It identifies a real limitation of prior AET-based methods: they depend on finite random sampling in the simplex and do not explicitly optimize the perturbation distribution.

2. The paper provides a useful theoretical view. The second-order analysis gives a plausible explanation for why the perturbation distribution matters more than individual sampled points, and it connects well to the design of the method.

3. The empirical results are solid overall. The full method is consistently better than its ablations and improves over SA-AET on the main retrieval benchmarks.

**Weaknesses**

1. The theoretical justification feels local. The key result is based on a second-order approximation under small perturbations, so it mainly supports the intuition rather than fully explaining the iterative attack pipeline in a highly nonlinear VLM setting.

2. The paper itself admits extra overhead from optimizing distribution parameters, and the appendix also notes higher cost when increasing the number of mixed candidates. But there is no clear compute comparison against the strongest baselines.

3. For MLLM evaluation, using KL divergence over output token distributions shows distribution shift, but it is still weaker than task-level evaluation of semantic failure or generation quality. Does this attack have similarities with the work [1,2]? If so, it can be discussed in the experiment.

[1] A Frustratingly Simple Yet Highly Effective Attack Baseline: Over 90% Success Rate Against the Strong Black-box Models of GPT-4.5/4o/o1

[2] VEAttack: Downstream-agnostic Vision Encoder Attack against Large Vision Language Models

---

> ### Author Rebuttal · Authors · 2026-03-30
>
> **W1**: Our theoretical derivation provides design intuition and local justification, rather than fully characterizes attack process in highly nonlinear VLMs. Although VLMs are nonlinear, adversarial attacks are performed via iterative small-step updates: *Each step operates locally around current input*. Under this regime, second-order approximation is reasonable and standard to *capture local behavior of loss landscape and motivate distribution-aware gradient aggregation*.
>
> Empirically, from **Tabel 5 in Appendix C.2**, as $\epsilon$ increases from small values, attack success rate of DDGA improves significantly. However, as $\epsilon$ becomes larger, performance gain turns to be *marginal*, due to the increasing approximation error of neglected higher-order terms.
>
> **W2**: For complexity, let $T$ be number of attack iterations, $V$ be number of augmentation views, $K$ be number of sampled points, $D$ be image dimension, and $d$ be Dirichlet dimension. Let $C_{f}$ and $C_{b}$ as forward and backword costs, and $C_{g}=C_{f}+C_{b}$ as total gradient cost. For strong baseline DRA and SA-AET, each iteration requires $KV$ gradient evaluations, leading to complexity ${\cal{O}}(TKVC_{g})$. In contrast, DDGA performs only one gradient evaluation per iteration, and replaces remaining operations with forward-only evaluations and lightweight distribution updates, resulting in complexity: ${\cal{O}}(T(KVC_{f}+C_{g}+d+KD))$, where $d$ and $KD$ correspond to lightweight policy update and calculation of covariance-guided matrix. Since $C_{g}>C_{f}$, DDGA reduces gradient computations for image attack.
>
> We report empirical runtime and average performance with source model CLIP_ViT on Flickr30K and MSCOCO:
>
> |Method|Time (s/epoch)|Complexity|Flickr30K|MSCOCO|
> |-|-|-|-|-|
> |SGA|4.80|${\cal{O}}(TVC_{g})$|38.82 $\pm$ 15.40|56.02 $\pm$ 13.71|
> |DRA|5.30|${\cal{O}}(TKVC_{g})$|46.25 $\pm$ 17.78|65.24 $\pm$ 13.76|
> |SA-AET|7.31|${\cal{O}}(TKVC_{g})$|52.74 $\pm$ 15.72|69.61 $\pm$ 12.82|
> |DDGA (ours)|8.27|${\cal{O}}(T(KVC_{f}+C_{g}+d+KD))$|55.98 $\pm$ 14.14 (3.24 $\uparrow$)|75.13 $\pm$ 10.71 (5.52 $\uparrow$)|
>
> Although DDGA reduces gradient computations, it introduces additional forward and lightweight distribution operations, thus slightly higher wall-clock time (modest overhead 8.27s vs 7.31s). However, it achieves improvements over SA-AET (**+3.24%** on Flickr30K and **+5.52%** on MSCOCO), demonstrating favorable efficiency–performance trade-off.
>
> **W3**: For semantic evaluations, we use Qwen3-Reranker-4B to measure retrieval similarity between original and generated texts, and employ GLM-4.7-Flash as LLM-judge to rank semantic relevance of generated texts (higher means lower relevance).
>
> |Method|Qwen3-Reranker-4B ⬇|GLM-4.7-Flash ⬆|
> |-|-|-|
> |DRA|0.22 $\pm$ 0.37|1.89 $\pm$ 0.75|
> |SA-AET|0.17 $\pm$ 0.34|1.95 $\pm$ 0.87|
> |DDGA (ours)|0.15 $\pm$ 0.32|2.16 $\pm$ 0.80|
>
> DDGA yields lower similarity scores (0.15 vs 0.17/0.22) and higher semantic rankings (2.16 vs 1.95/1.89), indicating stronger semantic disruption. More qualitative visualizations from Qwen3-VL-4B-Instruct, LLaVA-v1.6-Mistral-7B, and GPT-5.2 are in Figure 1-3 in https://anonymous.4open.science/r/icml8075-70DC/README.md.
>
> Since DDGA is **untargeted attack**  to make victim model produce incorrect outputs deviating from source semantics, we do not compare it with M-Attack [1], a **targeted attack** specifically matches semantics of target image (evaluated via target-conditioned KMRScore). VEAttack [2] assumes a **gray-box** setting with access to vision encoder, while DDGA follows a **transfer-based black-box** paradigm. Both attack objectives and threat models are different.
>
> **Q1**: To isolate contribution of image-side distribution learning, we remove text attack and compare against strong baselines under same setting.
>
> ||ALBEF||TCL||CLIP_CNN||
> |-|-|-|-|-|-|-|
> |Method|TR|IR|TR|IR|TR|IR|
> |DRA w/o TA|23.73|40.30|27.11|40.09|59.29|65.32|
> |SA-AET w/o TA|33.28|48.56|35.20|45.74|63.63|69.51|
> |DDGA w/o TA|36.77|51.90|39.21|50.05|66.82|72.10|
>
> DDGA outperforms DRA and SA-AET, indicating our gain mainly comes from the proposed image-side distribution learning. We also provide multiple evaluations using adversarial images: Table 3 (IC task) directly reflects image-side effectiveness, while MLLM results (Figure 3) and visualizations (Figure 5 in Appendix) demonstrate impact of image perturbations.
>
> **Q2**: To evaluate stability of policy optimization, we report cosine similarity of consecutive updates in Dirichlet parameters ($\Delta\alpha^{(t)}=\alpha^{(t)}-\alpha^{(t-1)}$) of all sampels. As shown, average similarity remains 0.6–0.7 across iterations, indicating policy updates are aligned.
>
> |step|1|2|3|4|5|6|7|8|
> |-|-|-|-|-|-|-|-|-|
> |$cos(\Delta\alpha^{(t-1)},\Delta\alpha^{(t)})$|0.598|0.687|0.655|0.687|0.631|0.667|0.655|0.656|
>
> More evidences on convergence and stability of policy optimization are in Figure 4 in https://anonymous.4open.science/r/icml8075-70DC/README.md.

---

> > ### Author Rebuttal · Reviewer_ascp · 2026-04-04
> >
> > Thank you for the response. My concerns have been adequately addressed.

---

> > > ### Author Response · Authors · 2026-04-06
> > >
> > > We sincerely thank you for the recognition of our work and your constructive feedback.
> > > We will carefully revise our manuscript based on the suggestions provided to further improve its quality and clarity.
> > > We greatly appreciate your time and effort.

---

### Official Review · Reviewer_nJXM · 2026-03-13

**Soundness:** 4
**Presentation:** 3
**Significance:** 3
**Originality:** 3
**Overall Recommendation:** 5
**Confidence:** 3

**Summary:**

This paper proposes DDGA (Dirichlet Distributional Gradient Aggregation), a transferable adversarial attack for vision-language models that replaces heuristic simplex sampling in AET-style methods with a learned Dirichlet mixing policy. The method optimizes the simplex mixing distribution via policy gradient and further uses the closed-form covariance of the learned Dirichlet distribution to construct orthogonal perturbations that increase gradient diversity. The paper evaluates DDGA on image-text retrieval, cross-task transfer to image captioning, and transfer to Qwen3-VL-4B-Instruct, reporting consistent improvements over prior baselines.

**Compliance With Llm Reviewing Policy:**

Affirmed.

**Key Questions For Authors:**

Please see the weaknesses above.

**Limitations:**

yes

**Strengths And Weaknesses:**

## Strengths



- The paper addresses a clear limitation of prior AET/DRA-style attacks. These methods rely on random sampling inside the adversarial simplex, which can be unstable and inefficient. DDGA instead learns a perturbation distribution directly. This is a clean and meaningful improvement.
- The method is well organized. The paper first motivates optimizing the expected perturbation, then introduces a Dirichlet policy for simplex mixing, and finally uses the covariance of the learned distribution for structured exploration.
- The experiments cover multiple datasets and settings, including Flickr30K and MSCOCO for retrieval, transfer to BLIP for captioning, and transfer to Qwen3-VL-4B-Instruct. The reported results show consistent gains in transferability.



---

## Weaknesses

1. Some of the target models are not very different from the surrogate models. In particular, the surrogate CLIPViT uses a ViT-B/16 image encoder, and the supplementary says that ALBEF and TCL also use ViT-B/16 image encoders. Because of this overlap, part of the reported transferability may come from similar visual backbones instead of stronger model-agnostic transfer. I would like to see more discussion, or more experiments, on targets with clearly different visual encoders and training setups. The Qwen3-VL result helps since it uses SigLIP2-Large, but that evaluation is also limited.

2. For Qwen3-VL-4B-Instruct, the evaluation only uses KL divergence between output token distributions under a fixed prompt. This shows that the output distribution changes, but it is less interpretable than the task-based metrics used in the other experiments. A semantic evaluation, such as LLM-as-a-judge or stronger qualitative examples, would make this result more convincing.

3. The paper says that DDGA adds only limited overhead, but a clearer runtime or cost comparison with DRA and SA-AET would strengthen the practical value of the method. The supplementary discusses the tradeoff between performance and computation, but a more direct comparison would still be helpful.

---

> ### Author Rebuttal · Authors · 2026-03-29
>
> **W1**: To address concern on backbone similarity, we extend evaluation to more diverse architectures, including CLIP ViT-L/14 and ALBEF, and perform cross-backbone transfer in multiple directions. Specifically, we use CLIP_ViT-L/14 as source model to evaluate transferability from totally different architecture ViT-L/14 to ViT-B/16 and CNN-based CLIP. Conversely, we also adopt ALBEF (ViT-B/16) as source model to measure transferability to larger-scale backbones such as ViT-L/14.
>
> ||Target|ALBEF||TCL||CLIP_CNN||CLIP_ViT-B/16||CLIP_ViT-L/14||
> |-|-|:-|:-|:-|:-|:-|:-|:-|:-|:-|:-|
> |Source|Method|TR|IR|TR|IR|TR|IR|TR|IR|TR|IR|
> |CLIP_ViT-L/14|DRA|23.98|38.45|25.92|39.74|53.38|60.62|60.12|67.46|100.00|99.93|
> ||SA-AET|30.34|44.13|31.19|44.17|55.43|63.02|66.63|73.00|100.00|99.57|
> ||DDGA (ours)|37.85|51.87|39.62|52.24|61.31|69.16|72.15|77.60|99.76|96.64|
> |ALBEF|DRA|99.90|99.98|91.57|91.17|49.55|59.01|46.26|56.80|30.66|49.85|
> ||SA-AET|99.90|99.95|96.42|96.02|57.22|65.59|55.58|63.89|34.75|53.52|
> ||DDGA (ours)|99.90|99.95|97.05|97.60|68.45|73.89|67.73|72.97|44.28|64.93|
>
> As shown, DDGA consistently improves over DRA and SA-AET across all settings, including heterogeneous transfers. These consistent gains indicate that improvement is not due to backbone overlap, but reflects stronger model-agnostic transferability. We will include these results in the revised version for clarity.
> On the other hand, while our current results on Qwen3-VL (SigLIP2-L) already indicate transferability to a different backbone, we should acknowledge that this evidence is still limited. We further extend this evaluation with more comprehensive analyses below (see W2) to strengthen the support.
>
> **W2**: We agree KL divergence alone is not sufficiently interpretable for evaluating MLLM outputs. To address this, we conduct additional semantic evaluations from both quantitative and qualitative perspectives. Specifically, we use Qwen3-Reranker-4B to measure **retrieval similarity** between original text and generated outputs (higher similarity indicates higher relevance), and employ GLM-4.7-Flash as an LLM-judge to rank the **semantic relevance** of generated texts to original text (higher rank indicates lower relevance).
>
> |Method|Qwen3-Reranker-4B ⬇️|GLM-4.7-Flash ⬆️|
> |-|-|-|
> |DRA|0.223 $\pm$ 0.371|1.89 $\pm$ 0.75|
> |SA-AET|0.172 $\pm$ 0.339|1.95 $\pm$ 0.87|
> |DDGA (ours)|0.149 $\pm$ 0.324|2.16 $\pm$ 0.80|
>
> As shown, DDGA consistently yields lower similarity scores (0.149 vs 0.172/0.223) and higher semantic rankings (2.16 vs 1.95/1.89), indicating stronger semantic disruption compared to DRA and SA-AET.
> Moreover, we provide qualitative visualizations of generated outputs from Qwen3-VL-4B-Instruct, LLaVA-v1.6-Mistral-7B, and GPT-5.2 in the anonymous link (Figure 1, 2, and 3 in https://anonymous.4open.science/r/icml8075-70DC/README.md), offering more evidence of generation quality degradation.
>
> **W3**: We agree that a clear complexity analysis is essential. For complexity, we denote $T$ as the number of attack iterations, $V$ as the number of augmentation views, $K$ as the number of sampled points, $D$ denote the image dimension, and $d$ denote the Dirichlet dimension. Let $C_{f}$ and $C_{b}$ as the forward and backword costs, and $C_{g}=C_{f}+C_{b}$ as the gradient costs. For strong baseline DRA and SA-AET, each iteration requires $KV$ gradient evaluations, leading to complexity ${\cal{O}}(TKVC_{g})$. In contrast, DDGA performs only one gradient evaluation per iteration, and replaces the remaining operations with forward-only evaluations and lightweight distribution updates, resulting in complexity: ${\cal{O}}(T(KVC_{f}+C_{g}+d+KD))$, where $d$ and $KD$ correspond to lightweight policy update and calculation of covariance-guided matrix. Since $C_{g}>C_{f}$, DDGA substantially reduces the number of expensive gradient computations for image-side attack.
>
> We also measure the empirical runtime and the average performance of each method with source model CLIP_ViT on both Flickr30K and MSCOCO:
>
> |Method|Time (s/epoch)|Complexity|Flickr30K|MSCOCO|
> |-|-|-|-|-|
> |SGA|4.799|${\cal{O}}(TVC_{g})$|38.82 $\pm$ 15.40|56.02 $\pm$ 13.71|
> |DRA|5.304|${\cal{O}}(TKVC_{g})$|46.25 $\pm$ 17.78|65.24 $\pm$ 13.76|
> |SA-AET|7.314|${\cal{O}}(TKVC_{g})$|52.74 $\pm$ 15.72|69.61 $\pm$ 12.82|
> |DDGA (ours)|8.273|${\cal{O}}(T(KVC_{f}+C_{g}+d+KD))$|55.98 $\pm$ 14.14 (3.24 $\uparrow$)|75.13 $\pm$ 10.71 (5.52 $\uparrow$)|
>
> Although DDGA reduces the number of expensive gradient computations, it introduces additional forward passes and lightweight distribution operations, leading to slightly higher wall-clock time in practice (see column 2).
> In practice, although DDGA introduces modest overhead (8.273s vs 7.314s), it achieves substantially better performance (**+3.24%** on Flickr30K and **+5.52%** on MSCOCO over SA-AET), demonstrating a favorable efficiency–performance trade-off.
> We will enhance this analysis in final version with clearer cost breakdowns and more detailed comparisons.

---

> > ### Author Rebuttal · Reviewer_nJXM · 2026-04-02
> >
> > Thank you for the response. It has adequately addressed my concerns.

---

> > > ### Author Response · Authors · 2026-04-06
> > >
> > > We sincerely thank you for the recognition of our work and your constructive feedback.
> > > We will carefully revise our manuscript based on the suggestions provided to further improve its quality and clarity.
> > > We greatly appreciate your time and effort.

---

### Decision · Program_Chairs · 2026-04-30

**Decision:**

Accept (regular)

**Comment:**

The paper proposes DDGA, a distribution-aware adversarial attack that learns a Dirichlet policy to optimize perturbation aggregation for improved transferability in vision-language models.

The method is technically sound, well-motivated, and a meaningful improvement over AET-style sampling, with clear empirical gains across tasks and models.
Reviewer's concerns focused on limited backbone diversity, interpretability of MLLM evaluation, and computational overhead; however, the authors provided substantial additional experiments (e.g., cross-backbone transfer, semantic evaluation, runtime analysis) that satisfactorily addressed these issues, and reviewers explicitly acknowledged resolution.

Overall, the contribution is solid, and the method is reproducible, and this paper is likely to be useful to the community.